# Long-term upper-troposphere climatology of potential contrail occurrence over the Paris area derived from radiosonde observations

Kevin Wolf[1], Nicolas Bellouin[1,2], and Olivier Boucher[1]

[1]Institut Pierre–Simon Laplace, Sorbonne Université / CNRS, Paris, France
[2]Department of Meteorology, University of Reading, Reading, United Kingdom

**Correspondence:** Kevin Wolf (kevin.wolf@ipsl.fr)

**Abstract.** Condensations trails (or contrails) that form behind aircraft have been of climatic interest for many years. Yet their radiative forcing is still uncertain. A number of studies estimate the radiative impact of contrails to be similar or even larger than that of $CO_2$ emitted by aviation. Hence, contrail mitigation may represent a significant opportunity to reduce the overall climate effect of aviation. Here we analyze an eight year data set of radiosonde observations from Trappes, France, in terms of the potential for contrail and induced cirrus formation. We focus on the contrail vertical and temporal distribution and test mitigation opportunities by changing flight altitudes and fuel type. Potential contrail formation is identified with the Schmidt–Appleman criterion (SAc). The uncertainty of the SAc, due to variations in aircraft type and age, is estimated by a sensitivity study and is found to be larger than the radiosonde measurement uncertainties. Linkages between potential contrail formation layers and the thermal tropopause as well as with the altitude of the jet stream maximum are determined. While non-persistent contrails form at the tropopause level and around 1.5 km above the jet stream, persistent contrails are located approximately 1.5 km below the thermal tropopause and at the altitude of the jet stream. The correlation between contrail formation layers and the thermal tropopause and jet stream maximum allows to use these quantities as proxies to identify potential contrail formation in numerical weather prediction models. The contrail mitigation potential is tested by varying today's flight altitude distribution. It is found that flying 0.8 km higher during winter and lowering flight altitude in summer reduces the probability for contrail formation. Furthermore, the effect of prospective jet engine developments and their influence on contrail formation are tested. An increase in propulsion efficiency leads to a general increase in the potential occurrence of non-persistent and persistent contrails. Finally, the impact of alternative fuels (ethanol, methane, and hydrogen) is estimated and found to generally increase the likelihood of non-persistent contrails and to a more limited extent persistent contrails.

## 1 Introduction

Global aviation significantly contributes to climate warming through a combination of factors. One of these factors is $CO_2$, with aviation being responsible for 2.5 to 2.6 % of the total anthropogenic $CO_2$ fossil fuel emissions in 2018 (Friedlingstein et al., 2019; Lee et al., 2021; Boucher et al., 2021). In addition to $CO_2$, the combustion of fossil fuels in jet engines releases nitrogen oxides ($NO_x$), sulfur dioxide ($SO_2$), sulfate, and water vapor among other by-products.

Of particular interest is water vapor emission as it allows to form condensation trails, also termed contrails, that emerge behind aircraft (Schumann, 1996; Kärcher, 2018). The emitted aerosol particles, which act as condensation nuclei, influence the contrail formation from excess water vapor in the exhaust plume. Most of the time these contrails vanish within a few seconds or minutes but they can be persistent up to a day depending on the environmental conditions (Jensen et al., 1994; Schumann, 1996; Haywood et al., 2009).

Whether a contrail can develop in the first place is usually estimated with the Schmidt–Appleman criterion (SAc, Schmidt, 1941; Appleman, 1953) on the basis of simple thermodynamic principles. The original SAc was revised and simplified by Schumann (1996) and slightly reformulated by Rap et al. (2010). The SAc defines a critical temperature $T_{\mathrm{crit}}$, above which contrails cannot form, and a critical relative humidity $RH_{\mathrm{crit}}$, which is a function of both the ambient and critical temperatures, above which contrails form. The critical temperature also depends on the ambient air pressure, aircraft–engine specific parameters, and fuel properties. When the ambient air fulfills the SAc, excess water vapor within the exhaust plume deposits on available particles in the exhaust plume (mostly soot) to form liquid droplets. Subsequently, the exhaust plume cools and the liquid water droplets freeze into ice crystals. The SAc does not differentiate between short-lived and persistent contrails. For contrails to be persistent the ambient air must be supersaturated with respect to ice in so called ice supersaturated regions (ISSR).

When ambient conditions favor persistent contrails, the contrails will undergo a transition from their line-shaped appearance, develop into larger clouds, mix, and merge with surrounding clouds depending on the vertical wind shear and entrainment rate (Unterstrasser and Stephan, 2020). Eventually, persistent contrails can transform into widespread contrail cirrus (Jensen et al., 1998; Haywood et al., 2009). Climate models and satellite observations suggest such contrail cirrus artificially increases the global cloud cover by 6 to 10 % in northern hemisphere mid-latitudes with consequential effects on global climate (Burkhardt and Kärcher, 2011; Quaas et al., 2021). However, it is still unclear to which extent contrails alter the occurrence of natural cirrus. Contrails modify the water vapor budget around them, leading to a competition for available water vapor supersaturation through condensation on ice particles (Ponater et al., 2021). This can lead to reduced natural cloud cover, a change in cloud optical properties, as well as in the lifetime of natural cirrus (Burkhardt and Kärcher, 2011; Ponater et al., 2021). Contrails and contrail cirrus are known to have a small cooling effect in the shortwave part of the radiation spectrum but a heating effect in the longwave part (Chen et al., 2000). The magnitude and dominance of the heating and cooling depends on multiple factors, including altitude of the cloud, optical thickness, solar zenith angle, underlying surface albedo, surface temperature (Meerkötter et al., 1999).

The radiative effect of a perturbation is quantified by its radiative forcing (RF), which is defined as the difference between the net irradiance at the tropopause with and without the presence of the perturbation being considered. While the RF for aviation-induced $CO_2$ is estimated to be ca. 30 mW m$^{-2}$ (Lee et al., 2021; Boucher et al., 2021), non-$CO_2$ effects may have similar or even larger RF (Burkhardt and Kärcher, 2011; Lee et al., 2021). In spite of intensive investigation within the last decade the actual RF by contrails and contrail cirrus remains uncertain. For young, linear contrails Burkhardt and Kärcher (2011) estimated a RF in the longwave of 5.5 mW m$^{-2}$ and a shortwave RF of $-1.2$ mW m$^{-2}$, leading to a net forcing around 4.3 mW m$^{-2}$. Burkhardt and Kärcher (2011) further estimated the RF of contrail-cirrus to be 47.1 mW m$^{-2}$ in the longwave

spectrum and $-9.6$ mW m$^{-2}$ in the shortwave wavelength range, with a net RF of 37.5 mW m$^{-2}$. A more recent study by
Bock and Burkhardt (2016), using an elaborate cloud model, estimated a RF of up to 106 mW m$^{-2}$ for 2006. In their review
paper Lee et al. (2021) determined a best estimate forcing of 57.4 mW m$^{-2}$ for 2018, with a 90 % likelihood range from 17
to 98 mW m$^{-2}$. The variety in estimated contrail RF highlights the importance to further investigate contrails with respect to
their distribution in time and space, their temporal evolution, and related radiative effects.

Non-CO$_2$ effects thus represent a large, yet uncertain, contribution of aviation to climate change for which mitigation options
may exist. This is of particular interest as a transition to potentially carbon-neutral fuels like ethanol, methane, or liquid
hydrogen will not prevent contrail formation (Gierens, 2021). Mitigation of contrails has been suggested as a potential solution,
and may be achieved by rerouting flights and/or changing the flight altitudes (Rosenow et al., 2018; Teoh et al., 2020a, b).
Despite the attractiveness of the idea, rerouting is challenging for many reasons. In particular it remains difficult to correctly
predict and parameterize contrails in climate and numerical weather prediction (NWP) models, particularly due to uncertainties
in relative humidity (Gierens et al., 2020). Therefore, reference observations are required to evaluate the performance of such
models and their ability to predict accurately contrails and their RF as a function of the flight path.

A global perspective on contrails can be obtained from satellites. For example, Meyer et al. (2002) used satellite observations
to determine the contrail RF at the regional scale. Iwabuchi et al. (2012) used a combination of lidar measurements from Cloud-
Aerosol Lidar and Infrared Pathfinder Satellite Observations (CALIPSO) and imagery from the Moderate Resolution Imaging
Spectroradiometer (MODIS) to determine the physical and optical properties of persistent contrails. More recent studies by
Schumann et al. (2021), Quaas et al. (2021), and Digby et al. (2021) investigated the influence of the flight restrictions in the
wake of the Covid19 pandemic to constrain the contribution of contrail cirrus to the total cirrus cloud coverage. Unfortunately,
the spatial resolution of most actual meteorological satellite instruments is limited to 250 m or more. Using passive remote
sensing in the thermal infrared wavelength range further decreases the resolution of 1 km or more. Therefore, it is likely that
most young contrails remain undetected by current meteorological satellites. Commercial high-resolution satellite imagers may
help but their revisit time is limited. Investigation of the early stages of contrail formation and transformation is thus proving to
be difficult using satellite data. Satellites also provide only a restricted vertical resolution that further complicates the derivation
of profiles.

An alternative to satellite remote sensing are in situ observations of cirrus and contrail clouds, whether they are obtained
during dedicated aircraft campaigns (e.g., Voigt et al., 2017; Bräuer et al., 2021) or from the long-term data set processed by
the In-service Aircraft for a Global Observing System (IAGOS; Petzold et al., 2015) . In addition to airborne observations,
radiosonde (RS) measurements can provide a good insight into the vertical profile of the atmosphere. RS measurements are
regularly performed in space and time for the sake of NWP, which allows to derive climatologies over long periods. Further-
more, they cover the entire vertical column with a relatively good vertical resolution. As a consequence, they are not limited to
the flight levels of the present-day fleet of aircraft as currently sampled by IAGOS. This is of particular interest when investi-
gating future impacts of an alternative fleet of aircraft that rely on alternative fuels (e.g., liquid hydrogen) and/or that operates
at a different altitude range. RS also have disadvantages. For instance they are limited in their payload and do not allow for
additional instrumentation like cloud particle counters. Furthermore, RS are subject to environmental conditions, particularly

low temperatures and insolation, which leads to increasing measurement uncertainties with altitude. Nevertheless, using basic post-processing techniques the influence of the environmental conditions on the measurements can be reduced and reliable observations can be retrieved.

Within the last two decades several studies have investigated the potential for contrail formation and its vertical distribution based on RS measurements. While Spichtinger et al. (2003) and Haywood et al. (2009) focused on a single station or a specific case study, Baughcum et al. (2009) used multiple stations in the US and, more recently, Agarwal et al. (2022) used data from the Integrated Global Radiosonde Archive (IGRA) and combined vertical profiles from a broad set of stations. Even though Agarwal et al. (2022) applied corrections on the RS profiles, not all RS types are generally capable of providing the required measurement accuracy to detect conditions prone to contrail formation and/or ISSR, especially at the colder temperatures.

Here we focus on the use of an eight year data set of RS performed by Météo–France from Trappes, France. The radiosonde station is located close to the Site Instrumental de Recherche par Télédétection Atmosphérique (SIRTA, Haeffelin et al., 2005), which is equipped with a set of passive and active remote sensing instruments that will complement the RS profiles in future work. Most importantly, a single type of RS (Meteomodem type M10) were launched throughout the selected eight year time period, and these RS measurements are currently being incorporated into the GCOS Reference Upper-Air Network (GRUAN Dirksen et al., 2014). The GRUAN measurement requirements are such that RS passing the GRUAN test are suitable for ISSR layer detection. Furthermore, with SIRTA being located in Western Europe, it is strongly affected by air traffic between Europe and America and is therefore representative of European flight traffic.

This study aims to derive vertical profiles of the potential for contrail formation and growth. We also seek relationships between contrail formation and two atmospheric features, namely the thermal tropopause and the jet stream.

The common separation in non-persistent and persistent contrail based on the SAc is extended to define a "reservoir" for potential contrail spreading. The reservoir is characterized by atmospheric conditions that are not favorable for persistent contrail formation but are nevertheless supersaturated with respect to ice and below the critical temperature $T_{\mathrm{crit}}$. Such atmospheric conditions are of particular interest because they are thermodynamically (and potentially spatially) close to regions where persistent contrails can form either because of colder temperature or larger RH. Contrails could therefore spread into such a regions through mixing on the vertical or horizontal direction.

This study also goes beyond Spichtinger et al. (2003), Haywood et al. (2009) and Agarwal et al. (2022) by investigating the role of alternative fuels on potential contrail formation and potential mitigation by flight altitude changes. By varying the flight altitude distribution (FAD) we quantify the potential of vertically shifting flights to reduce contrails.

Following this introduction, we present the utilized data for this study in Section 2 and the statistical processing methods to flag the contrail formation in Section 3. Section 4 describes the results of this study, with specific measures that are relevant for flight planning and trajectory optimization with regard to contrail mitigation. Finally, Section 5 summarizes the results and concludes. Appendix A provides a detailed explanation of the radiosonde post-processing.

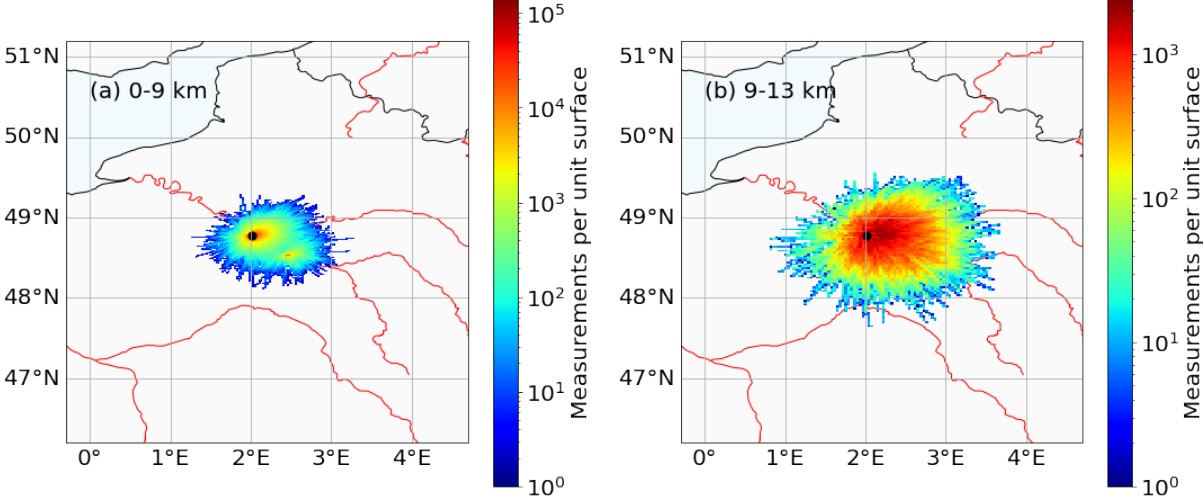

**Figure 1.** Spatial distribution of individual radiosonde observations around the Trappes station site ($48.77°\,\mathrm{N}, 2.01°\,\mathrm{E}$, filled black circle) of the full eight year period. The distributions are shown for measurements at altitudes (a) below 9 km and (b) between 9 and 15 km. The frequency of occurrence (measurements per box of $0.1° \times 0.1°$) is indicated by two different logarithmic color bars. The underlying map was created with the cartopy library (Met Office, 2010 - 2015).

## 2    Data

This study uses routine radiosonde (RS) launches made by Météo–France close to the city of Trappes, France. In follow up studies, these observations will be complemented and combined by observations from the Site Instrumental de Recherche par Télédétection Atmosphérique (SIRTA, Haeffelin et al., 2005). The SIRTA facility is located in Palaiseau, approximately 30 km

away from Trappes, and is equipped with an extensive set of passive and active remote sensing instruments, such as an all-sky camera to track contrail development, radiometers, and a lidar.

### 2.1    Radiosonde observations

We analyze an eight year data set of RS observations spanning the years 2012 to 2019. The spatial coverage and representativity of a RS station is determined by the distribution of wind direction and wind speed. Figure 1a–b shows frequency of occurrence

of all RS measurements of the analyzed period separated for altitudes below 9 km and between 9 and 13 km. The average horizontal advection at 11 km altitude of an RS ascent is approximately 85 km. Occasionally, horizontal displacements of up to 200 km are possible. The distributions are characterized by an elliptical shape with the major axis in the east-west direction, following the mean westerly flow at this location. Even though we analyze RS from a single station, the observations span a significant part of northern France, which is subject to intense flight traffic.

The measurements are performed with the M10 radiosonde from Meteomodem. The M10 radiosonde measures temperature with a thermistor-type sensor with an uncertainty of $\pm 0.22$ K below 20 km altitude (Dupont et al., 2021). The sensor is protected

with an aluminum coating, reflecting 95 % of the incoming shortwave and longwave radiation. Therefore, it is assumed that the measured temperature $T_{RS}$ is a good approximation of the ambient air temperature. Relative humidity (RH) is measured with a capacitor-type humidity sensor with an uncertainty of $\pm 3$ % (Dupont et al., 2021). The response time of the RH sensor is 2 s at 20°C and increases to 90 s at $-60$°C (Dupont et al., 2021).

RS measurements are subject to biases due in particular to the time lag of the sensor, chemical contamination of the RH sensor, and artificial heating from direct sunlight (Miloshevich et al., 2004). During daytime, direct sunlight can heat the exposed temperature and humidity sensors. Therefore, a post-processing of the radiosonde humidity measurements ($RH_{RS,liq}$) is mandatory to obtain reliable RH profiles. We have thus corrected the RS measurements for radiative heating and time lag of the RH sensor prior to our analysis. The applied corrections are detailed and evaluated in Appendix A.

For quality control, RS that do not reach a minimum altitude of 15 km and that contain spurious measurements, i.e., incomplete profiles and nonphysical temperature measurements, are screened out of the data set, which leaves 5512 full profiles out of a total of 5773 (or 95 %). Subsequent to the applied RH correction and removal of spurious data, the RS profiles are interpolated on a uniform vertical grid ranging from 0 to 18 km with a vertical resolution of 25 m.

RS measurements of RH ($RH_{RS,liq}$) are commonly defined with respect to a plane, liquid water surface. This is true even under conditions of supersaturation with respect to liquid water or ice (Nagel et al., 2001; Spichtinger et al., 2003). To identify ice supersaturation, $RH_{RS,liq}$ is converted to RH with respect to ice $RH_{RS,ice}$ following:

$$RH_{RS,ice} = RH_{RS,liq} \cdot \frac{e_{sat,liq}(T)}{e_{sat,ice}(T)} \tag{1}$$

with $e_{sat,liq}(T)$ and $e_{sat,ice}(T)$ the saturation water vapor pressures over liquid water and ice, respectively. The used equations and their validity ranges are given in Appendix B.

## 2.2 Flight altitude distributions

SIRTA is equipped with an Automatic Dependent Surveillance–Broadcast (ADS–B) receiver. These receivers record ADS-B signals that are periodically broadcast by the majority of commercial aircraft. The signals contain the aircraft latitude and longitude, altitude, and call sign. Depending on the aircraft altitude and atmospheric conditions, these ADS-B signals can be received from a distance of approximately 50 to 300 km around SIRTA.

We derive the vertical distribution of flight altitudes using the recorded ADS-B signals. Flight altitudes in ADS-B data are given in terms of flight levels (FL) expressed in feet but correspond in fact to pressure levels at cruising altitudes. The FL are converted to a "true" geometric altitude above the ground using daily surface pressure and temperature profile from the RS measurement.

The flight altitude distribution (FAD) includes all flights between 9 and 15 km which we take as representative of cruising altitudes. The data are binned into vertical intervals of 250 m. ADS-B data from 2019 are used to compute monthly distributions. However no distinct seasonal cycle in the FAD was detected so we only consider an annual mean distribution in the rest of this study. Air traffic regulations force aircraft to follow specific flight levels, which leads to a FAD that exhibit discrete

**Table 1.** Specific energy $Q$ and emission index $EI$ of kerosene (Jet-A1), ethanol, methane, and hydrogen. Values of specific energy $Q$ and energy density are from Schumann (1996) and NIST (2022).

| Fuel | Unit | Kerosene | Ethanol | Liquid methane | Liquid hydrogen |
|---|---|---|---|---|---|
| Specific energy $Q$ | $\mathrm{MJ\,kg^{-1}}$ | 43.2 | 27.2 | 50 | 120 |
| Energy density | $\mathrm{MJ\,l^{-1}}$ | 34.9 | 21.6 | 21.2 | 8.4 |
| Mass water vapor emitted per unit energy | $\mathrm{kg\,MJ^{-1}}$ | 0.026 | 0.043 | 0.045 | 0.075 |
| Emission index of water vapor $EI_{H_2O}$ | $\mathrm{kg\,kg^{-1}}$ | 1.25 | 1.17 | 2.25 | 8.94 |
| Ratio of $EI_{H_2O}$ to that of kerosene | - | 1 | 0.64 | 1.8 | 7.15 |

layers. Given that contrails get mixed and diluted in the atmosphere, the annual mean FAD is smoothed with a boxcar filter and a window of two layers. The FAD is normalized so as to obtain a probability density function (PDF), $p_{\mathrm{FA}}(z)$, of the flight altitude over the SIRTA ($\int_z p_{\mathrm{FA}}(z)\,\mathrm{d}z = 1$).

## 3 Methods

### 3.1 Flagging of contrails and ISSR in the RS data

For contrail to form, the ambient air must be sufficiently cold and moist. Appleman (1953) estimated critical threshold temperatures and RH based on thermodynamic principles. This neglects the complicated dynamics that occur in the jet and vortex phases of a contrail but has proved to be a valid first-order approximation.

Within this study, RS measurements of temperature and RH are used to determine the potential occurrence of non-persistent contrails (NPC) and persistent contrails (PC). The detection is based on the revised SAc by Schumann (1996), which was slightly reformulated by Rap et al. (2010). Borrowing the notations of Rap et al. (2010), the threshold temperature $T_{\mathrm{crit}}$ (in K), above which no contrail can form, is approximated by:

$$T_{\mathrm{crit}} = 226.69 + 9.43 \cdot \ln\left(G - 0.053\right) + 0.72 \cdot \ln^2\left(G - 0.053\right), \tag{2}$$

with $G$ the slope (in $\mathrm{Pa\,K^{-1}}$) of the water vapor pressure-temperature relationship in the engine exhaust plume as it gets diluted in the ambient air. Specifically, $G$ is determined as:

$$G = \frac{EI_{H_2O} \cdot c_{\mathrm{p}} \cdot p}{\epsilon \cdot Q \cdot (1 - \eta)}, \tag{3}$$

where $Q$ is the specific combustion heat of the fuel (in $\mathrm{J\,kg^{-1}}$), $EI$ the emission index of water vapor for the fuel (in $\mathrm{kg\,kg^{-1}}$), $\eta$ the propulsion efficiency of the aircraft, $c_{\mathrm{p}} = 1004\,\mathrm{J\,kg^{-1}\,K^{-1}}$ the isobaric heat capacity of air, $p$ the ambient air pressure (in Pa) of the flight, and $\epsilon \approx 0.622$ the ratio of the molecular weights of water vapor and dry air. Values for $Q$ and $EI$ for kerosene

(Jet-A1), ethanol, methane, and hydrogen are given in Table 1. Finally, the critical relative humidity $RH_{\text{crit}}$ is determined by:

$$RH_{\text{crit}}(T) = \frac{G \cdot (T - T_{\text{crit}}) + e_{\text{sat}}^{\text{liq}}(T_{\text{crit}})}{e_{\text{sat}}^{\text{liq}}(T)}, \tag{4}$$

with $T$ the ambient air temperature and $e_{\text{sat}}^{\text{liq}}(T)$ and $e_{\text{sat}}^{\text{liq}}(T_{\text{crit}})$ the saturation water vapor pressures at their respective temperatures. For modern aircraft–engine combinations a propulsion efficiency of $\eta = 0.3$ is assumed (Rap et al., 2010). According to Schumann (2000), $\eta$ is defined by:

$$\eta = \frac{F \cdot v}{\dot{m}_{\text{f}} \cdot Q}, \tag{5}$$

which is the ratio between the work rate $F \cdot v$ and the amount of energy released $\dot{m}_{\text{f}} \cdot Q$ during the combustion process. As $\eta$ is

200 also a function of the aircraft speed $v$, which depends on the aerodynamics of the aircraft, $\eta$ must be interpreted as a parameter that depends on the combination of aircraft and engine.

With the definition of $T_{\text{crit}}$ and $RH_{\text{crit}}$ from Eqs. 2–5, the water-vapor-pressure–temperature diagram, shown in Fig. 2, can be separated into four areas below the $e_{\text{sat}}^{\text{liq}}$ curve. Region 1 (R1) meets the critical thresholds of the SAc but is unsaturated with respect to ice, hence, it indicates the potential for R1-NPC. Region 2 (R2) represents atmospheric conditions fulfilling

the SAc that are also ice supersaturated. This region indicates the potential for R2-PC. Region 3 (R3) includes conditions not fulfilling the SAc but that are ice supersaturated. While conditions for contrail formation are not met in R3, this region can be understood as a potential reservoir, where contrails (R3-R) formed nearby can also spread and persist through mixing. The basis for considering R3 is that contrails can transition into widespread cirrus in both the R2-PC and R3-R regions. We can schematically understand the contrail spreading into the R3-R region both on the horizontal and vertical direction on the

diagram of Fig. 2. Moving along the water vapor saturation pressure line for a constant temperature (i.e., parallel to the water vapor pressure axis) can be understood as contrail spreading on a (more or less horizontal) isotherm surface. This is plausible as the water vapor field is known to vary on short spatial scales. Similarly moving along a temperature line for a constant relative humidity (i.e., parallel to the temperature axis) can be understood as contrail spreading on the vertical. This interpretation remains qualitative as there is no guarantee that data points from the R2-PC and R3-R regions that are close to each other on

the diagram are also close to each other in the atmosphere. Nevertheless we believe it is an interesting addition to the usual interpretation of the R1-NPC and R2-PC regions. To the authors best knowledge, the occurrence of such R3-R conditions has not been quantified before. Finally, for completeness, we consider a fourth region R0, which corresponds to the complement to the union of R1-NPC, R2-PC, and R3-R so that any point in the diagram belongs to either R0, R1-NPC, R2-PC, or R3-R.

Table 2 provides an overview of the criteria for R1-NPC, R2-PC, and R3-R. It is once again pointed out that in our study

the SAc does not directly indicate contrail formation but instead flags the potential for NPC and PC. In the following we have processed the data in order to flag each layer (geometric thickness $\Delta z = 25$ m) of the individual RS profile as belonging to one of the four regions. This results in a probability function $P_z(\text{Rx})$, defined at each altitude $z$, with four discrete values such that:

$$P_z(\text{Rx} = \text{R0}) + P_z(\text{Rx} = \text{R1} - \text{NPC}) + P_z(\text{Rx} = \text{R2} - \text{PC})$$
$$+ P_z(\text{Rx} = \text{R3} - \text{R}) = 1. \tag{6}$$

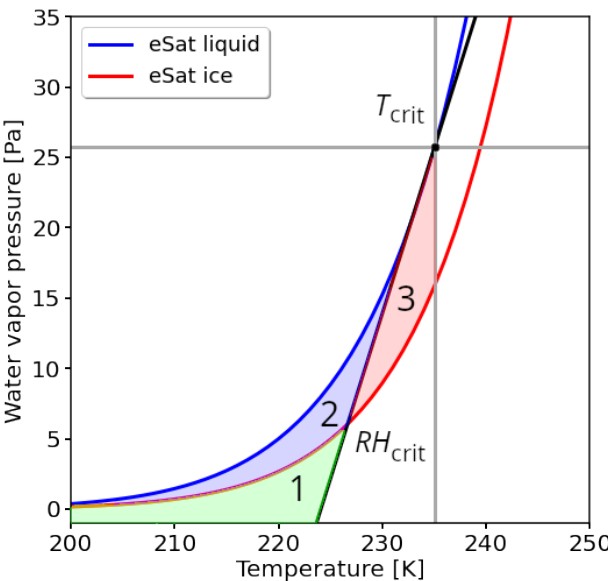

**Figure 2.** Water-vapor-pressure–temperature diagram with saturation water vapor pressure over ice (red curve) and liquid water (blue curve). Conditions prone to the formation of NPSs are shown in green (R1-NPC); conditions prone to the formation of PC are shown in blue (R2-PC). The potential reservoir for spreading contrail is highlighted in red (R3-R). The critical temperature and relative humidity determined by the Schmidt–Appleman criterion are located on the black line, which separates potential contrail formation (left) from no contrail formation (right).

**Table 2.** Separation of the three regions of potential contrail formation in the water-vapor-pressure–temperature diagram. The regions are defined by temperature $T$ and relative humidity over liquid $RH_{\mathbf{liq}}$ or ice $RH_{\mathbf{ice}}$. Index "crit" identifies critical values from the Schmidt–Appleman criterion (SAc).

| Region | $T$ | $RH_{\mathrm{liq}}$ | $RH_{\mathrm{ice}}$ | SAc | ISSR | Characteristic |
|--------|-----|---------------------|---------------------|-----|------|----------------|
| 1 | $T < T_{\mathrm{crit}}$ | $RH_{\mathrm{liq}} > RH_{\mathrm{crit,liq}}$ | $RH_{\mathrm{ice}} < 1$ | ✓ | ✗ | non-persistent |
| 2 | $T < T_{\mathrm{crit}}$ | $RH_{\mathrm{liq}} > RH_{\mathrm{crit,liq}}$ | $RH_{\mathrm{ice}} > 1$ | ✓ | ✓ | persistent |
| 3 | $T < T_{\mathrm{crit}}$ | $RH_{\mathrm{liq}} < RH_{\mathrm{crit,liq}}$ | $RH_{\mathrm{ice}} > 1$ | ✗ | ✓ | spreading |

## 3.2 Sensitivity of $T_{\mathbf{crit}}$ and $RH_{\mathbf{crit}}$ on $\eta$

A key parameter in calculating $T_{\mathrm{crit}}$ and $RH_{\mathrm{crit}}$ is the propulsion efficiency $\eta$. For modern aircraft, like the Airbus A380 with a Rolls–Royce Trent 900 engine, $\eta$ is approximately 0.3, a commonly applied value in contrail studies (Schumann, 2000; Rap et al., 2010). This value is a best guess but future jet engines might become more efficient, which leads to an increased $\eta$.

Furthermore, the variety of aircraft models, engine types, and engine ages leads to variations in the aircraft-engine specific $\eta$,
introducing an uncertainty in the calculated $T_{\mathrm{crit}}$ and $RH_{\mathrm{crit}}$. Therefore, the sensitivities of $T_{\mathrm{crit}}$, $RH_{\mathrm{crit}}$, and related potential
contrail formation on $\eta$ have to be studied.

To determine the sensitivity of potential contrail formation, $\eta$ is varied between 0.25 and 0.40 with increments of 0.05. $T_{\mathrm{crit}}$
and $RH_{\mathrm{crit}}$ profiles are calculated using the ambient temperature $T$ from the US standard atmosphere profile. Figure 3 shows
profiles of $T_{\mathrm{crit}}$, $RH_{\mathrm{crit}}$, and their respective absolute differences for the variation in $\eta$. The general increase in $T_{\mathrm{crit}}$ with
increasing $\eta$ comes with more efficient engines (larger $\eta$), as these are characterized by colder exhaust plumes and, hence,
contrails form at higher ambient temperatures. For an increase (decrease) in $\eta$ of 0.05, the absolute difference in $T_{\mathrm{crit}}$ is almost
constant over all altitude layers with a increase (decrease) in $T_{\mathrm{crit}}$ of around 0.8 K. For relative humidity, absolute differences
in $RH_{\mathrm{crit}}$ below 10 km altitude are smaller than 5 %. Above an altitude of 10 km the differences in $RH_{\mathrm{crit}}$ grow quicker with
altitude. At 12 km altitude $RH_{\mathrm{crit}}$ decreases (increases) by around 25 % due to an increase (decrease) in $\eta$ of 0.05.

The measurement uncertainty of $RH_{\mathrm{RS,liq}}$ from the radiosonde humidity sensor is reported to be smaller than 3 %. For the
GRUAN-compliant M10 post-processing an uncertainty of 1.21 % is assumed. In this study, not all GRUAN corrections were
applicable and we do use a conservative measurement uncertainty of 5 %, mostly arising from uncertainties in the temperature
profile (see Appendix A). This leads to a total, maximal uncertainty of 9 % in $RH_{\mathrm{RS}}$, which is below or equal to the uncertainty
on $RH_{\mathrm{crit}}$ due to $\eta$. Therefore, uncertainties in $RH_{\mathrm{RS}}$, due to uncertainties in $T_{\mathrm{crit}}$ and $RH_{\mathrm{crit}}$, are smaller than the variation
in $RH_{\mathrm{crit}}$, which relaxes the constrains on the required accuracy of the RS observations. Consequently, we argue here that the
RS measurements, even with only basic corrections for $T$ and $r$, can be used together with the SAc to detect potential contrail
formation.

### 3.3  Joint probabilities of contrail occurrence and flight altitude distribution

To estimate the actual contrail formation caused by air traffic, the frequency and vertical position of flight tracks have to be
considered as well. Treating the two events (Rx conditions and flight altitude) as independent, we multiply the probabilities,
$P_z(\mathrm{Rx})$, from Section 3.1, with the flight altitude PDF, $p_{\mathrm{FA}}(z)$, from Section 2.2:

$$p(\mathrm{Rx}, z) = P_z(\mathrm{Rx}) \cdot p_{\mathrm{FA}}(z), \tag{7}$$

with Rx taking the values R0, R1-NPC, R2-PC, or R3-R. Technically, $p(\mathrm{Rx}, z)$ is a joint PDF that has the peculiarity of
depending on two random variables, one (Rx) being discrete and the other (altitude $z$) being continuous. By construction, the
joint PDF is normalized to 1:

$$\sum_{\mathrm{R} \in \{\mathrm{R0, R1-NPC, R2-PC, R3-R}\}} \int_z p(\mathrm{Rx} = \mathrm{R}, z) \, dz = 1. \tag{8}$$

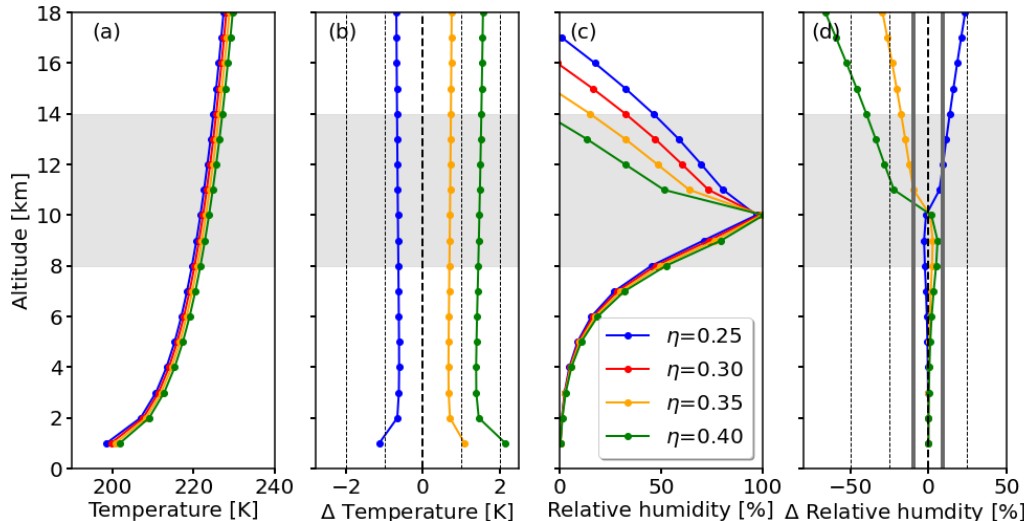

**Figure 3. (a,c)** Vertical profiles of critical temperature $T_{\mathrm{crit}}$ and $RH_{\mathrm{crit}}$ for different values of the propulsion efficiency $\eta$ ranging from 0.25 to 0.40. **(b,d)** Absolute differences in $T_{\mathrm{crit}}$ and $RH_{\mathrm{crit}}$ with respect to their values for $\eta = 0.3$. Uncertainties from the radiosonde observations are estimated to be 9 % and are indicated by the two vertical solid, gray lines on panel d). The gray shaded area indicates the altitudes of major interest for potential contrail formation.

### 3.4 Identification of thermal tropopause and jet stream location

The World Meteorological Organization (WMO) defines the location of the TT layer based on the the lapse rate, $\gamma$, of the vertical temperature profile:

$$\gamma = \frac{dT}{dz},\tag{9}$$

The thermal tropopause is located at the lowest level at which $\gamma$ decreases to $-2\,\mathrm{K\,km^{-1}}$ or below (in absolute value) and the average value of the overlying 2 km of the atmosphere is not smaller than $-2\,\mathrm{K\,km^{-1}}$ (WMO, 1957). For each RS profile, $\gamma$ is calculated with Eq. 9 and the location of the smallest $\gamma$ (in absolute value), i.e., the local minimum in $T$, between 8 and 14 km is set as the TT. For profiles, where the altitude of smallest $\gamma$ was equal to 8 or 14 km, the TT altitude was not identifiable and was removed from the analysis.

The derived wind measurements of RS observations are used to identify the vertical position of the maximum wind speed within the profile. We consider the RS measurements to be within the jet stream if the wind speed exceeds $30\,\mathrm{m\,s^{-1}}$ at some location on the vertical (Gibbs and Newton, 1958). Otherwise the profile of the day is rejected and not used to calculate the vertical distribution.

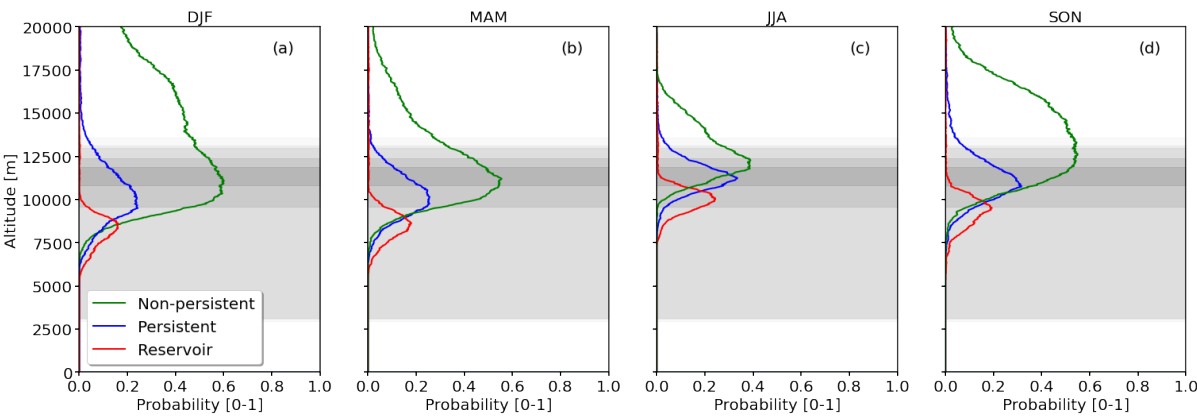

**Figure 4.** Probabilities, $P_z(\text{Rx})$, of meeting R1-NPC (green curve), R2-PC (blue curve), and R3-reservoir conditions (red curve) shown as a function of altitude. The four panels show the averages for DJF, MAM, JJA, and SON seasons. The flight altitude distribution over the SIRTA is provided by the gray shading indicating the 10, 25, 50, 75, and 90[th] percentiles.

## 4 Results

### 4.1 Frequency of contrail formation and ISSR from radiosonde

Seasonally-averaged vertical distributions of frequency of occurrence of the R1-NPC, R2-PC, R3-R conditions (see Fig. 2) are calculated on the basis of the individual flagged RS profiles and shown in Figs. 4a–d. Generally speaking, all seasons are dominated by R1-NPC conditions (NPC, green curve) with the highest frequency of occurrence and the largest vertical extent throughout the year. During the winter months (Fig. 4a), the probability to form R1-NPC reaches 60 % at altitudes between 10 and 11 km. Also R1-NPC has the largest vertical extent, with an altitude range from 8 km to well above an altitude of ca. 15 km, above which the uncertainties of the radiosonde measurements are of the same magnitude than variations in $RH_{\text{crit}}$ due to variations in $\eta$. During the summer months, R1-NPC conditions show a minimum occurrence with a peak of 39 % between 11 and 12 km altitude and a lower overall occurrence frequency. Spring and autumn are considered as transition seasons showing intermediate values of potential formation with maxima on the vertical of 56 % and 54 %, respectively.

Generally lower frequencies of occurrence are detected for R2-PC conditions (PC, blue curve) with peak values of 33 % during summer and 24 % in spring. Autumn is characterized by an intermediate peak probability of 31 % and for winter a maximum of around 28 % is identified. The annual cycle in R2-PC conditions is less pronounced compared to the R1-NPC conditions. The largest vertically-integrated occurrence is in winter followed by spring, autumn, and summer, similarly to R1-NPC conditions. This is comparable to Petzold et al. (2020) who analyzed five years of Measurement of Ozone and Water Vapor by Airbus In-Service Aircraft (MOZAIC Marenco et al., 1998) observations. Petzold et al. (2020) found a minimum in ISSR, a prerequisite for R2-PC, of 20 % to 30 % in summer and 35 % to 40 % in winter. Nevertheless, Petzold et al. (2020) observed a stronger seasonal dependency compared to our analysis. In contrast, much lower occurrences were found by Rädel and Shine (2007), with only 10 % of the winter profiles and 17 % of the summer profiles containing R2-PC conditions. The

difference between our study and Rädel and Shine (2007) might be explained by the filtering of the radiosonde data. Rädel and Shine (2007) used only measurements below the tropopause and rejected profiles with $RH_{\text{liq}}$ larger than 90 % to remove conditions with iced sensors. The filtering might bias the results towards lower $RH$ profiles. Even lower frequency of occurrence are given for by a recent paper from Agarwal et al. (2022), who found R2-PC conditions in only 3 % to 6 % of the profiles launched between 30°N and 60°N. The vertical position of the peaks of the R2-PC conditions are located between 10 and 11.5 km, which is around one kilometer higher compared to Rädel and Shine (2007), who estimated the mean altitude of ISSR between 9.5 and 10 km in winter and summer, respectively. Similar altitudes like in the present paper are reported by Agarwal et al. (2022), with around 9 km in winter and 11 km in summer.

The maximum frequencies for R3-R conditions (potential contrail spreading region) range from 16 % in winter to 24 % in summer. Figures 4a–d clearly show that R3-R conditions tend to be located at lower altitudes than R2-PC conditions, which is consistent with the fact that the R3-R corresponds to larger air temperature than the R2-PC region on Fig. 2. Assuming that R2-PC and R3-R conditions coexist on the vertical, this implies that persistent contrails formed under R2-PC conditions that descend to lower altitudes can persist and spread under R3-R conditions. This could significantly increase the volume or lifetime of persistent contrails. From the RS profiles it is estimated that 80 % of the profiles that were flagged for R2-PC conditions were also flagged for R3-R conditions somewhere in the vertical below R2-PC.

The differences in frequency of occurrence and mean altitude of R2-PC conditions between this study, Petzold et al. (2020), and Agarwal et al. (2022) likely result from different sampling regions. While the present study uses radiosondes from one station, Petzold et al. (2020) uses observations that cover Europe, the North Atlantic, and North America, and Agarwal et al. (2022) uses selected radiosonde observations between 30°N and 60°N. Furthermore, the reliability and accuracy of radiosonde observations near and above the tropopause are difficult and require careful post-processing. The methods and filtering during the post-processing process can influence the results and explain the differences between our analysis, Rädel and Shine (2007), and Agarwal et al. (2022).

## 4.2   Connections between potential contrail formation and the thermal tropopause

As pointed out in Section 3.1, certain criteria have to be fulfilled to initiate contrail formation and persistence. Characteristic features of the atmosphere like the thermal tropopause (TT) or the location of the jet stream might favor or disfavor the occurrence of NPC and PC. Furthermore, these characteristic features are well resolved in general circulation models (GCM) for climate or numerical weather prediction (NWP) while small scale processes on RH at the sub-grid level are more challenging to predict. Therefore, the TT and the jet stream might be suitable proxies or predictors for diagnosing and predicting contrail occurrence from observations and models.

As an example, using 15 months of RS observations from Lindenberg, Spichtinger et al. (2003) found that ice-supersaturation frequently occurs close to the lower boundary of the TT. Similarly, Diao et al. (2015) analyzed aircraft observations and identified that most of the ISSR appear ±500 m around the TT. By definition the TT is associated with the lowest temperature. The frequent occurrence of ISSR close to the TT is caused by the inhibition of vertical mixing. The TT also suppresses humidity exchange with the stratosphere (Petzold et al., 2020). Therefore, advected humid air from lower altitudes, for instance along

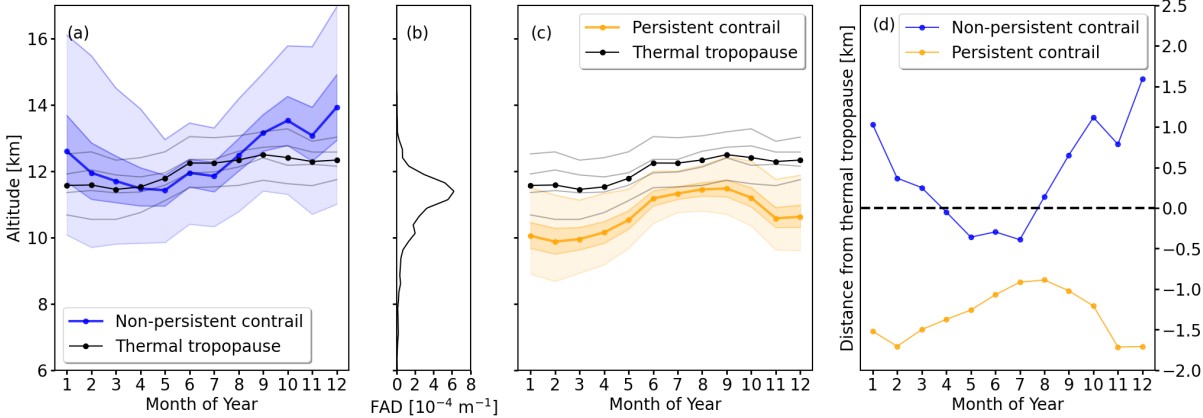

**Figure 5. (a)** Seasonal cycle of the vertical distribution of the altitudes of potential NPC with median altitudes (solid, blue line) and the 20, 40, 60, and 80[th] percentiles (shaded areas). The mean altitude of the TT is given in black and the 20, 40, 60, and 80[th] percentiles are indicated by gray lines. **(b)** Vertical flight altitude distribution from ADS-B data. **(c)** Same as (a) but for PC (solid orange lines and shaded areas). **(d)** Relative distance between the median NPC and PC altitudes and the median TT altitude.

warm-conveyor belts, is likely to aggregate just below the TT. The combination of low temperatures, adiabatic cooling, and
325 enhanced humidity is thus favorable for ice cloud formation (Eguchi and Shiotani, 2004; Kim et al., 2016).

Figure 5a shows the median altitude of the TT in relation to the normalized vertical distribution of R1-NPC. (It should be noted that the vertical distribution of the R1-NPC distribution is cut at 20 km, which has an impact on the computation of the percentiles during the winter season as R1-NPC can be formed higher according to Fig. 4a). The TT is lowest in January with a median altitude of 11.5 km. The median altitude is highest in September with 12.3 km. Between February and September the
330 median altitude of R1-NPC is above the TT, while for the remainder of the year the TT is located below. Figure 5d shows the relative distance between the median altitude of potential R1-NPC and the TT. The largest distance between R1-NPC and TT appears in December with the R1-NPC 1.5 km above the TT. During summer, the R1-NPC is 0.3 km below the TT.

Similarly, the location of the PC relative to the TT is shown in Fig. 5c. Thorough the entire year the R2-PC is located below the TT and follows the annual distribution of the TT. The largest relative distance is found during winter with values of 1.6 km
below the TT. R2-PC is closest to the TT in summer, particularly in August with a location 1 km below the TT. This is in line with observations from Spichtinger et al. (2003), who detected ISSR between 0 and 2.5 km below the TT. Similar observations were made by Petzold et al. (2020), who used IAGOS aircraft data to find the respective locations of TT and PC.

Figure 5b shows the FAD derived from ADS-B data. It is noteworthy that the median of the R1-NPC overlaps with the FAD peak from March to June. Consequently, any kind of flight activity during this time of the year and at these altitudes is likely
to cause some kind of R1-NPC formation. It has also to be mentioned that flying above the TT is associated with contrail formation in the lower stratosphere (LS). Contrails within the LS are prone to extended lifetimes due to stronger stratification of ambient air and weaker dilution (Schumann et al., 2017). To avoid the formation of R1-NPC contrails, the region 1.5 km

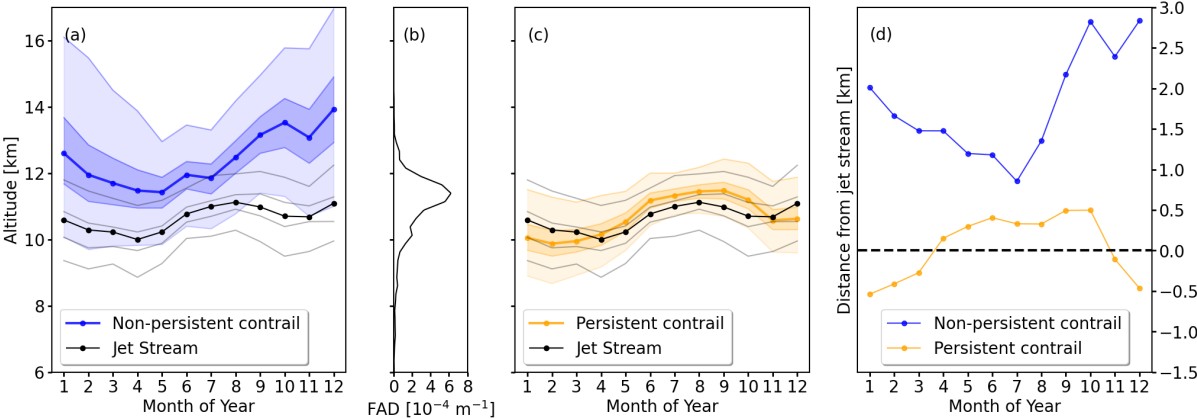

**Figure 6. (a)** Seasonal cycle of the vertical distribution of the altitudes of potential R1-NPC with median altitudes (solid, blue line) and the 20, 40, 60, and 80[th] percentiles (shaded areas). The distributions of R1-NPC and R2-PC are limited to the upper bound of 20 km. The mean altitude of the jet stream is given in black and the 20, 40, 60, and 80[th] percentiles are indicated by gray lines. **(b)** Vertical flight altitude distribution from ADS-B data. **(c)** Same as panel (a) but for R1-NPC (solid, orange line and shaded areas). **(d)** Relative distance between the median R1-NPC and R2-PC altitudes and the median jet stream altitude.

below the TT should be avoided during March through June as the chance for formation of R2-PC is largest. Flying lower is feasible as the R2-PC region is mainly 1.5 km below the TT.

### 4.3 Connections between potential contrail formation and the jet stream

Like the TT, the jet stream is a trackable feature of the mid-latitude atmosphere and often used by aviation on eastbound flights across the Atlantic. The jet stream, with its high wind speeds and wind shear, is known to cause upper level divergence and convergence, depending on location and curvature of the wind field. Divergence occurring close to the tropopause is associated with raising air masses, which are adiabatically cooled and advect humidity from lower levels. The combination of humidity and low temperatures favors the formation of R1-NPC and R2-PC. Irvine et al. (2012) found that the location of ISSR correlates with high wind speeds and, in particular, with the jet stream position.

Figure 6a shows the RS-based median jet stream altitude as a function of height. The lowest altitude of the jet stream is detected in April at 10 km and reaches a maximum in August with 11 km, leading to an annual median variation of 1 km. The distance between R1-NPC and the jet stream ranges from 0.8 km (July) to 2.8 km (September) above the jet stream (Fig. 6d, blue line).

Similarly, Fig. 6c visualizes the vertical distribution for R2-PC. The analysis clearly shows that R2-PC form closer to the jet stream than R1-NPC. The smallest distances are identified for April and November with the jet stream at the same altitude as for R2-PC. In winter, R2-PC is located 0.5 km below the jet stream and from April to October R2-PC are up to 0.5 km above the jet stream.

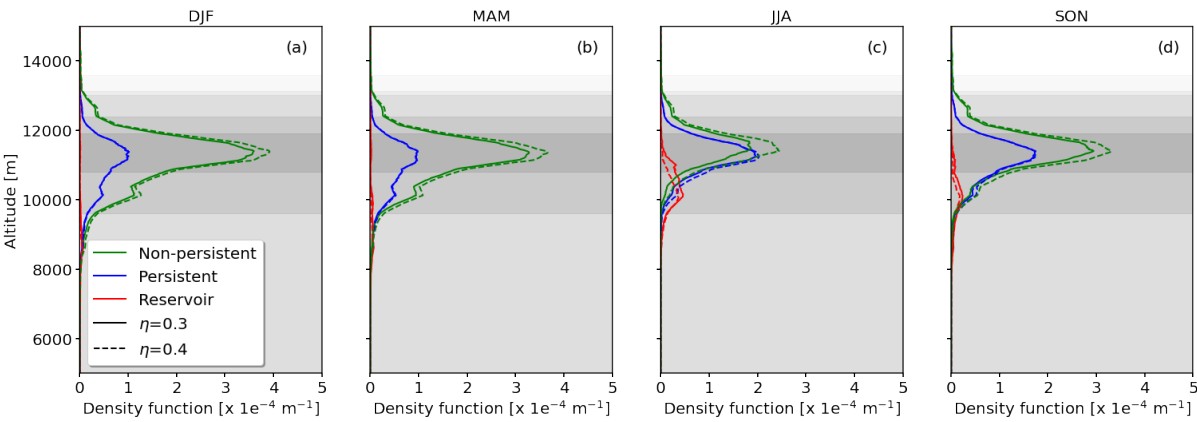

**Figure 7.** Joint probability, $p(\mathrm{Rx} = \mathrm{R}, z)$, of an aircraft flying through a region that satisfies the conditions for R1-NPC (green curves), R2-PC (blue curves) or R3-R at a given altitude. The FAD is indicated by the gray shadings that show the 10, 25, 50, 75, and 90[th] percentiles. The four panels represent the four seasons. The effect of the change of the propulsion efficiency $\eta$ is illustrated by the solid and dashed lines for $\eta = 0.3$ and $\eta = 0.4$, respectively.

Based on the distribution given in Fig. 6c, flying above the jet stream from January to May and below the jet stream for the months of June to October, can reduce R2-PC formation.

## 4.4 FAD weighted contrail occurrence

Weighting the vertical distributions of region R1-NPC to R3-R with the actual FAD leads to the joint probability of an aircraft flying through layers that meet either of these conditions. The distributions are weighted with Eq. 7 (Sect. 3.3). The joint
PDF is shown in Fig. 7a–d as a set of three curves representing $p(\mathrm{Rx} = \mathrm{R}, z)$ for the three values of R of interest. Each of the curve can be interpreted as the PDF for a flight to meet one of the conditions given the current PDF of flight altitude over SIRTA. It should be noted that $p(\mathrm{Rx} = \mathrm{R}, z)$ is not normalized to unity but rather to the probability of a flight to meet a given set of atmospheric conditions. The PDF for R1-NPC formation remains the largest in all seasons except for JJA at lower altitudes. Indeed $p(\mathrm{Rx} = \mathrm{R1} - \mathrm{NPC}, z)$ reaches a maximum of $3.6 \cdot 10^{-4}$ m$^{-1}$ in winter, followed by spring and autumn with
$3.3 \cdot 10^{-4}$ m$^{-1}$ and $2.9 \cdot 10^{-4}$ m$^{-1}$, respectively. The minimum is smallest for the summer months at around $1.9 \cdot 10^{-4}$ m$^{-1}$. The PDF for R2-PC formation, $p(\mathrm{Rx} = \mathrm{R2} - \mathrm{PC}, z)$, is usually less than that for R1-NPC formation, except for JJA, and tends to peak at a little lower altitude. This leads to a maximum of $p(\mathrm{Rx} = \mathrm{R2} - \mathrm{PC}, z)$ during summer at $2.0 \cdot 10^{-4}$ m$^{-1}$, exceeding the otherwise dominating value for R1-NPC. Autumn follows with a probability of $1.7 \cdot 10^{-4}$ m$^{-1}$. For spring and winter the same probability of $1.0 \cdot 10^{-4}$ m$^{-1}$ is determined. The reservoir plays a marginal role due to the location at the lower end of the
FAD. During winter and spring, the reservoir is almost not existing. Peak probabilities of $p(\mathrm{Rx} = \mathrm{R3} - \mathrm{R}, z)$ of $0.5 \cdot 10^{-4}$ m$^{-1}$ and $0.2 \cdot 10^{-4}$ m$^{-1}$ are calculated for summer and autumn, respectively. Nevertheless, contrails that potentially formed within R2-PC could spread in this region.

The vertically integrated values of contrail-weighted R2-PC occurrence ranges between 15 % and 22 %. This is significantly higher compared to Agarwal et al. (2022), who identified only 3 % to 5 % of the fuel-burn weighted profiles as suitable for persistent contrails. Potential explanations for the differences are the data sets for weighting the profiles and the overall lower occurrence of PC in the Agarwal et al. (2022) analysis.

The differences in frequency of occurrence and mean altitude of R2-PC conditions between this study, Petzold et al. (2020), and Agarwal et al. (2022) likely result from the deviating sampling regions. While the present study uses radiosondes from one station, Petzold et al. (2020) uses observations that cover Europe, the North Atlantic, and North America, and Agarwal et al. (2022) uses selected radiosonde observations between 30°N and 60°N. Furthermore, the reliability and accuracy of radiosonde observations near and above the tropopause are difficult and require careful post-processing. The methods and filtering during the post-processing process can influence the results and explain the differences among our analysis, Rädel and Shine (2007), and Agarwal et al. (2022).

Table 3 provides a summary of the modal altitudes and peak values of the probabilities, $p(\mathrm{Rx} = \mathrm{R}, z)$, for the different seasons. In addition, we computed the vertically-integrated marginal probabilities:

$$P_{\mathrm{int}}(\mathrm{Rx} = \mathrm{R}) = \int_z p(\mathrm{Rx} = \mathrm{R}, z)\, \mathrm{d}z, \tag{10}$$

which represents the probability of a flight to meet one of the R1-NPC, R2-PC, or R3-R conditions over the RS site, or in other words, the traffic-weighted probability of meeting the R1-NPC, R2-PC, or R3-R conditions.

## 4.5 Influence of the propulsion efficiency on the occurrence of non-persistent and persistent contrail

Aircraft engines might become even more efficient in the future, which leads to an increase in the propulsion efficiency $\eta$. Larger $\eta$ are achieved through increased work done with the same amount of fuel, which results in reduced heat energy remaining in the exhaust plume. A cooler plume, albeit with a similar amount of humidity, will result in a larger $G$ and $T_{\mathrm{crit}}$ prone to contrail formation. Previous studies, like one from Schumann (2000), showed that an increase in $\eta$ leads to enhanced contrail formation. As a proxy for a future scenario, we investigate the effect of $\eta = 0.4$ on the likelihood of contrail formation.

Figures 7a–d show the air traffic-weighted vertical PDFs for regions R1-NPC, R2-PC, and R3-R for $\eta = 0.3$ (solid line) and $\eta = 0.4$ (dashed line). Comparing the two distributions, the largest effect in changing $\eta$ appears for R1-NPC. The increase in $\eta$ leads to higher R1-NPC formation especially in summer, with the maximum value in $p(\mathrm{Rx} = \mathrm{R1} - \mathrm{NPC}, z)$ increasing from $1.9 \cdot 10^{-4}\,\mathrm{m}^{-1}$ to $2.4 \cdot 10^{-4}\,\mathrm{m}^{-1}$ (+26 %). A similar, but slightly lower increase in the peak value is detected for the other seasons of the year.

In contrast, R2-PC is only slightly affected by the change in $\eta$. Similarly, no change in R3-R is identified for winter and spring. Only in summer and autumn, the increase in $\eta$ reduces the chance for R3-R occurrence. In these seasons R1-NPC and R2-PC occurrence is low in the first place. A certain fraction of R3-R just missed the requirements of the SAc. Increasing $\eta$ is related to colder exhaust plumes, shifting the line in Fig. 2 and causes the transition form R3-R into either of the contrail formation regions R1-NPC or R2-PC.

**Table 3.** Modal altitudes and peak values for the $p(\mathrm{Rx} = \mathrm{R}, z)$ PDFs for R=R1-NPC, R2-PC, and R3-R and for the four seasons. Vertically-integrated marginal probabilities are also given. Relative differences (in %) given in parentheses are calculated with respect to $\eta = 0.3$.

| Propulsion efficiency $\eta$ | | Winter | Spring | Summer | Autumn |
|---|---|---|---|---|---|
| 0.3 | Altitude mode R1-NPC [km] | 11.4 | 11.4 | 11.5 | 11.4 |
| | Altitude mode R2-PC [km] | 11.4 | 11 | 11.4 | 11.3 |
| | Altitude mode R3-R [km] | 8.6 | 8.6 | 10.2 | 10.1 |
| | Peak value R1-NPC [x $10^{-4}$ m$^{-1}$] | 3.6 | 3.3 | 1.9 | 2.9 |
| | Peak value R2-PC [x $10^{-4}$ m$^{-1}$] | 1.0 | 1.0 | 2.0 | 1.7 |
| | Peak value R3-R [x $10^{-4}$ m$^{-1}$] | 0.1 | 0.1 | 0.5 | 0.2 |
| | $P_{\mathrm{int}}(\mathrm{Rx} = \mathrm{R1} - \mathrm{NPC})$ | 0.514 | 0.445 | 0.21 | 0.363 |
| | $P_{\mathrm{int}}(\mathrm{Rx} = \mathrm{R2} - \mathrm{PC})$ | 0.156 | 0.153 | 0.208 | 0.220 |
| | $P_{\mathrm{int}}(\mathrm{Rx} = \mathrm{R3} - \mathrm{R})$ | 0.009 | 0.015 | 0.061 | 0.034 |
| 0.4 | Altitude mode [km] R1-NPC | 11.4 | 11.4 | 11.4 | 11.4 |
| | Altitude mode [km] R2-PC | 11.4 | 11.4 | 11.2 | 11.3 |
| | Altitude mode [km] R3-R | 8.6 | 8.6 | 10.2 | 10.1 |
| | Amplitude R1-NPC [x $10^{-4}$ m$^{-1}$] | 3.9 (9 %) | 3.7 (12 %) | 2.4 (26 %) | 3.3 (14 %) |
| | Amplitude R2-PC [x $10^{-4}$ m$^{-1}$] | 1.0 (0 %) | 1.0 (0 %) | 2.0 (0 %) | 1.8 (6 %) |
| | Amplitude R3-R [x $10^{-4}$ m$^{-1}$] | 0.1 (0 %) | 0.1 (0 %) | 0.4 (−20 %) | 0.2 (0 %) |
| | $P_{\mathrm{int}}(\mathrm{Rx} = \mathrm{R1} - \mathrm{NPC})$ | 0.568 (11 %) | 0.505 (14 %) | 0.283 (35 %) | 0.426 (17 %) |
| | $P_{\mathrm{int}}(\mathrm{Rx} = \mathrm{R2} - \mathrm{PC})$ | 0.159 (2 %) | 0.158 (4 %) | 0.229 (10 %) | 0.232 (6 %) |
| | $P_{\mathrm{int}}(\mathrm{Rx} = \mathrm{R3} - \mathrm{R})$ | 0.004 (−30 %) | 0.01 (−34 %) | 0.045 (−27 %) | 0.024 (−39 %) |

A summary of the peak values, respective altitudes, and vertically-integrated probabilities for the two $\eta$ values are given in Table 3.

## 4.6 Influence of selected fuels on contrail occurrence

The aviation industry is considering the transition from fossil fuels to alternative fuels like bio-ethanol, liquid methane, or hydrogen. Such fuels have the potential to reduce the overall aviation-induced $CO_2$ emissions, when generated from carbon-neutral sources.

We test the impact of three alternative fuels on contrail occurrence, namely ethanol, liquid hydrogen and liquid methane (see Table 1). We assume a constant aircraft-engine propulsion efficiency $\eta = 0.3$ and the same flight altitude distribution as for present-day conditions. In reality, one would expect $\eta$ and the flight altitude to vary if an aircraft was designed to use an alternative fuel and it may be interesting in the future to combine the expected changes. It should also be noted that the transition to hydrogen- or ethanol-powered jet engines is unlikely in the short term. However, flight tests with mixtures of

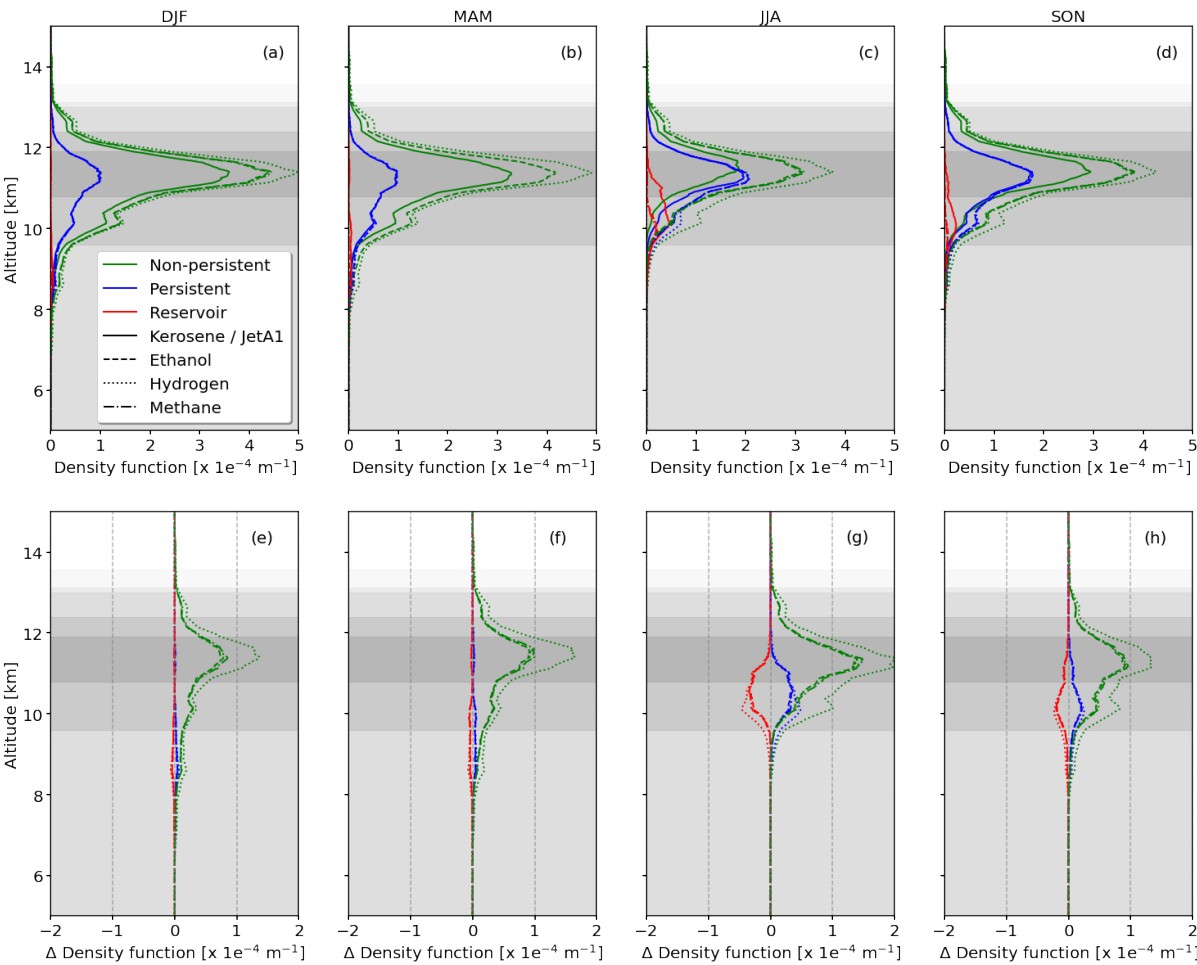

**Figure 8. (a–d)** Joint probability, $p(\mathrm{Rx} = \mathrm{R}, z)$, of an aircraft flying through a region that satisfies the conditions for R1-NPC (green curves), R2-PC (blue curves) or R3-R at a given altitude. The flight altitude distribution is provided by the gray shading indicating the 10, 25, 50, 75, and 90[th] percentile. PDFs are shown for kerosene / Jet-A1 (solid), ethanol (dashed), hydrogen (dotted), and methane (long dashed). The four panels represent the four seasons. **(e–h)** Absolute differences of the PDFs with respect to kerosene / Jet-A1, given by the black, dash-dotted line.

**Table 4.** Vertically-integrated air traffic-weighted probabilities for R1-NPC, R2-PC, and R3-R conditions. The results are provided for four fuel types (Jet-A1, ethanol, methane, and hydrogen) assuming a propulsion efficiency $\eta = 0.3$. Relative differences (in %) are given in parentheses with respect to Jet-A1.

| Season | Region | Jet-A1 / Kerosene | Ethanol | Methane | Hydrogen |
|--------|--------|-------------------|---------|---------|----------|
| Winter | $P_{\text{int}}(\text{Rx} = \text{R1} - \text{NPC})$ | 0.514 | 0.640 (25 %) | 0.650 (27 %) | 0.728 (42 %) |
|        | $P_{\text{int}}(\text{Rx} = \text{R2} - \text{PC})$ | 0.156 | 0.163 (5 %) | 0.163 (5 %) | 0.165 (6 %) |
|        | $P_{\text{int}}(\text{Rx} = \text{R2} - \text{PC})$ | 0.009 | 0.003 (−70 %) | 0.002 (−76 %) | 0 (−100 %) |
| Spring | $P_{\text{int}}(\text{Rx} = \text{R1} - \text{NPC})$ | 0.445 | 0.595 (34 %) | 0.607 (37 %) | 0.704 (58 %) |
|        | $P_{\text{int}}(\text{Rx} = \text{R2} - \text{PC})$ | 0.153 | 0.164 (7 %) | 0.165 (8 %) | 0.168 (10 %) |
|        | $P_{\text{int}}(\text{Rx} = \text{R3} - \text{R})$ | 0.015 | 0.004 (−75 %) | 0.003 (−80 %) | 0 (−100 %) |
| Summer | $P_{\text{int}}(\text{Rx} = \text{R1} - \text{NPC})$ | 0.209 | 0.392 (87 %) | 0.406 (94 %) | 0.535 (155 %) |
|        | $P_{\text{int}}(\text{Rx} = \text{R2} - \text{PC})$ | 0.208 | 0.253 (22 %) | 0.256 (23 %) | 0.274 (31 %) |
|        | $P_{\text{int}}(\text{Rx} = \text{R3} - \text{R})$ | 0.061 | 0.021 (−66 %) | 0.018 (−70 %) | 0.004 (−100 %) |
| Autumn | $P_{\text{int}}(\text{Rx} = \text{R1} - \text{NPC})$ | 0.363 | 0.512 (42 %) | 0.523 (44 %) | 0.612 (69 %) |
|        | $P_{\text{int}}(\text{Rx} = \text{R2} - \text{PC})$ | 0.220 | 0.246 (12 %) | 0.248 (13 %) | 0.256 (16 %) |
|        | $P_{\text{int}}(\text{Rx} = \text{R3} - \text{R})$ | 0.034 | 0.010 (−70 %) | 0.008 (−76 %) | 0 (−100 %) |

kerosene and ethanol are already underway. Here we calculate the PDFs of R1-NPC, R2-PC, or R3-R conditions for pure fuels and not for mixtures. Consequently, the vertical PDFs of R1-NPC to R3-R derived for different fuels provide the maximum effect by switching entirely to one of the alternatives. Fuel mixtures will lead to intermediate values of the vertical distributions depending on the stoichiometric mixture.

425 Each fuel, depending on its chemical structure, is characterized by the specific energy $Q$, the energy density, and the index of water vapor emission $EI_{\text{H2O}}$. Both, $Q$ and $EI_{\text{H2O}}$, are parameters in the SAc to determine $T_{\text{crit}}$ and $RH_{\text{crit}}$ for contrail formation (see Eq. 3). While $T_{\text{crit}}$ is only slightly affected by fuel types, particularly the offset of the tangent (black line in (Fig. 2), a change in fuel type will alter the slope and thus primarily determine the required ambient saturation.

 Subsequently, relative differences of the vertically-integrated distributions with respect to Jet-A1/kerosene are discussed.
Figure 8 shows that for the majority of the profiles (except the reservoir) the absolute difference is positive and, hence, switching to alternative fuels makes contrail formation more likely. For all seasons the largest increase is identified for NPC (green). While ethanol and methane lead to a similar increase in occurrence between +25 % in winter and up to +94 % in summer, the transition to hydrogen has the largest effect with an increase in NPC of +155 % in summer and the smallest difference of +42 % in winter.

 For winter and spring the relative changes in R2-PC are negligible ranging from +5 % to +10 %. During summer and autumn,
however, there is an increase in R2-PC between +12 % and +31 %. Simultaneously, the size of the reservoir region approaches zero. The transition from R3-R to R2-PC conditions is similar for all three fuel types due to the weighting with the FAD. Over

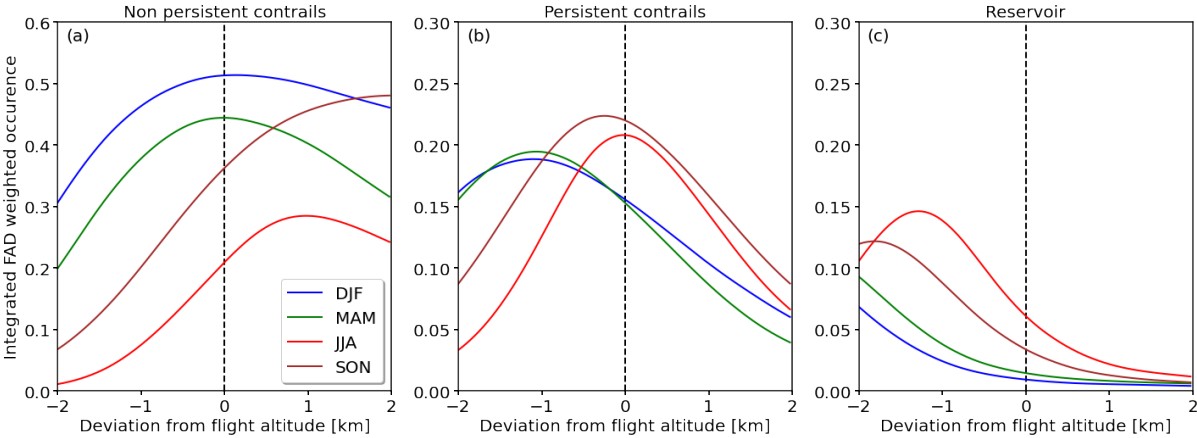

**Figure 9.** Vertically-integrated marginal probabilities $P_{\text{int}}(\text{Rx} = \text{R})$ for **(a)** R1-NPC formation, **(b)** R2-PC formation and **(c)** R3-R conditions as a function of the deviation in the FAD. Seasons are color-coded. The vertical dashed line at 0 km deviation represents the current median FAD of 11.3 km.

all, the largest changes in the vertically-integrated probability are identified for summer, which indicates that this season is most susceptible to a fuel change with respect to contrail formation. Contrarily, the smallest impact is determined for winter when the occurrence of R1-NPC and R2-PC is largest in the first place.

In the previous analysis one important aspect of contrail and cirrus formation is neglected. Switching to alternative fuel types is accompanied with a suggested reduction of the soot particle number concentration in the exhaust plume and the number of available ice condensation nuclei. Kärcher (2018) showed that for soot-rich regimes ($N \approx 10^{14} - 10^{16}\,\text{kg}^{-1}$), an increase in the particle number concentration leads to a linear increase in the number of activated ice crystals. Unexpectedly, a transition to a soot-poor regime ($N \approx 10^{12} - 10^{14}\,\text{kg}^{-1}$) leads to an increase of activated ice condensation particles. The increase in
the soot-poor regime is explained by the formation and activation of aqueaous particles, which subsequently freeze, when the ambient air is sufficiently cold and humid (Kärcher, 2018) . Case studies from Kärcher et al. (2015); Kärcher and Voigt (2017) as well as measurements from Moore et al. (2017) and Voigt et al. (2021) provide support to the model from Kärcher (2018) although there remain large uncertainties. In addition, a recent study by Bier et al. (2022) showed that switching to bio-fuels also modifies the size of the ice particles, which further modifies the optical properties, hence the radiative forcing, and the
contrail life-time.

### 4.7    Sensitivity of contrail formation to flight altitude

In this section, we investigate further how the probabilities of a flight to meet one of the R1-NPC, R2-PC, or R3-R conditions vary with an upward or a downward shift in the FAD. The current FAD is shown with the gray shadings Fig. 7 and is mostly concentrated between 9 and 14 km. We select a maximum shift of $\pm 2$ km around the median flight altitude (11.3 km), which
is assumed to remain close to the range of optimal aircraft operation. Figure 9a–c shows seasonal $P_{\text{int}}(\text{Rx} = \text{R})$ as a function

of the deviation from today's median flight altitude. Interestingly the probabilities $P_{\text{int}}(\text{Rx} = \text{R1} - \text{NPC})$ for R1-NPC formation ((Fig. 9a) have a maximum located at the current FAD median for winter and spring. Shifting the FAD to higher or lower altitudes generally would reduce the probability of forming R1-NPC, with a more effective reduction for decreasing FAD. Nevertheless, considering the probabilities $P_{\text{int}}(\text{Rx} = \text{R2} - \text{PC})$ for R2-PC formation (Fig. 9b), shifting flights to higher altitudes would reduce the likelihood of R1-NPC and R2-PC formation simultaneously. A different pattern is identified for the autumn and summer seasons. With $P_{\text{int}}(\text{Rx} = \text{R2} - \text{PC})$ having a local maximum at 0 km deviation and $P_{\text{int}}(\text{Rx} = \text{R1} - \text{NPC})$ decreasing towards lower altitudes, a downward shift would reduce R1-NPC and R2-PC formation at the same time.

As shown in Fig. 5 the region for R2-PC is subject to an annual cycle, which is highest in summer and lowest during the winter month. In summer the overlap with today's FAD is largest, with the median of R2-PC and FAD at similar altitude. Therefore, flying at higher or lower altitudes reduces the vertically integrated chance for R2-PC formation. During the winter months, when the R2-PC is generally lower, a shift to higher flight altitudes reduces the overlap. Similarly, the lower median altitude of R2-PC during winter explains why higher cruising altitudes reduce the vertically integrated chance for R2-PC.

The potential for a flight to cross the reservoir R3-R region is shown in Fig. 9c. With the reservoir always being located at the lower boundary of the R2-PC distribution (see Fig. 4), a shift of the FAD towards lower flight levels generally increases the likelihood to cross the reservoir, particularly in summer and autumn. In case of cloud- and contrail-free conditions, water vapor emitted by the aircraft within the reservoir region does not have an effect. However, if there are existing contrails or cirrus, the additionally emitted water vapor can deposit on available ice particles and sustains or fosters potential pre-existing clouds. Releasing additional water vapor under cloudy R3-R conditions might enhance cloud life-time as well as cloud optical and geometric thickness. At some point, the increase in ice water content and cloud optical thickness might compensate the longwave heating effect of the contrail and turn the net heating into a net cooling, which could be actively applied in flight planning. Nevertheless, whether such an intensified contrail has a warming or cooling effect depends on solar zenith angle, cloud microphysics, surface albedo, and surface temperature and cannot be estimated from RS observations alone. A detailed evaluation based on LES coupled with radiative transfer simulations is required. Furthermore, any cooling effect will vanish after sunset.

For completeness, it has to be mentioned that a transition towards lower flight altitude is bounded by the increase in air density and aerodynamic drag, leading to a disproportionate increase in fuel consumption. Still, toady's FAD result from a multi-variable optimization, which does not take into account the trade-off between additional $CO_2$ and potential contrail mitigation. There is also an upper altitude bound which depends on the aircraft characteristic. Furthermore, flying higher may have other drawbacks, especially if the fraction of flights cruising in the stratosphere increases. Emission of water vapor in the lower stratosphere has a stronger radiative effect compared to emissions in the tropopause. In addition, Gierens et al. (1999) found that, on some occasions, supersaturation can be present in the lower most part of the stratosphere and contrails that form in the lower stratosphere may have a longer life-time given the larger stratification compared to the troposphere (Gierens et al., 1999; Irvine et al., 2012). While supersaturation in the stratosphere might be rare, the increase in total flights in the lower stratosphere combined with the extended life-time can lead to an increased contrail coverage. It is also important to recognize that these results may not generalize to other sites and to other latitudes. A similar study would be needed with a much larger

set of RS locations before robust conclusions can be reached. One has also to consider that aircraft tend to fly at or close to their optimal flight levels. Flying lower or higher may be sub-optimal in terms of fuel consumption and/or require adjusting the aircraft airspeed. Furthermore, depending on an aircraft characteristics and payload, it may not always be possible to fly higher. Beyond that the interactions between flight altitude, aircraft performance, engine emissions, and radiative impact of possible contrails are complex. To estimate the effect of flight altitude changes dedicated studies have been performed, for example by Frömming et al. (2012), Dahlmann et al. (2016), and, more recently, by Matthes et al. (2021).

## 5 Summary

Condensation trails (or contrails) that form behind aircraft are estimated to have a similar radiative forcing (RF) as the $CO_2$ emitted by aviation (Lee et al., 2021). Therefore, the prospect of mitigating contrail formation and persistence is of high interest. A tentative solution that is getting some traction consists in actively rerouting a fraction of the flights to avoid atmospheric regions which are prone to persistent contrail formation. Flying around such regions requires the accurate forecast of their occurrence in time and space. Until today, numerical weather prediction and climate models suffer from large uncertainties in their representation of relative humidity and ice supersaturation.

We analyzed an eight year data set of radiosonde (RS) observations launched from Trappes, France. The RS are corrected for humidity-dry bias and time lag of the RH sensor. Using the Schmidt–Appleman criterion (SAc) and the ice-supersaturation threshold, the available RS profiles were flagged for their potential to host non-persistent contrails (R1-NPC) and persistent contrails (R2-PC). We introduced a third category, labeled as "reservoir", which does not fulfill the SAc but is nevertheless ice supersaturated. This reservoir provides an estimate for the potential spreading of existing contrails beyond the regions prone to R1-NPC and R2-PC formation.

Classification in R1-NPC and R2-PC with the SAc depends, among other parameters, on the propulsion efficiency $\eta$, which itself is a function of the actual aircraft-engine combination, aircraft type, and aircraft age, with typical values ranging between 0.2 and 0.35. Commonly, values of $\eta = 0.3$ are used but variations of $\pm 0.05$ are possible. We estimated the influence of variations in $\eta = 0.3 \pm 0.05$ on the thresholds of temperature $T_{\mathrm{crit}}$ and relative humidity $RH_{\mathrm{crit}}$ by applying the SAc to the US standard atmosphere. Increasing (decreasing) $\eta$ by 0.05 leads to a vertically constant increase (decrease) in $T_{\mathrm{crit}}$ by 0.8 K. An altitude dependence is found for $RH_{\mathrm{crit}}$, with continuously increasing values above 10 km altitude. At 14 km altitude a maximum increase in $RH_{\mathrm{crit}}$ of 12 % is identified. Within the altitude range of 8 to 14 km the variation in $T_{\mathrm{crit}}$ and $RH_{\mathrm{crit}}$ are larger than the measurement uncertainty of the corrected RS humidity profiles. Hence, we argue that the corrected RS are suited to identify potential contrail formation layers.

Labeling the individual RS measurements for R1-NPC, R2-PC, and the R3-R category, respective seasonal profiles of frequency of occurrence were derived. All seasons are dominated by the potential for R1-NPC with frequencies of around 60 %. R1-NPC are subject to a seasonal dependence with a maximum in winter and a minimum during summer. R2-PC are identified in 30 to 40 % of the profiles also with a seasonal dependence in altitude and occurrence frequency. The reservoir category is found in only $\approx 20$ % of the profiles.

Weighting the contrail formation potential with flight altitude distributions (FAD) derived from a colocated Automatic Dependent Surveillance–Broadcast (ADS–B) receiver provides vertical distribution probabilities for actual R1-NPC and R2-PC occurrence. The resulting profiles are still dominated by R1-NPC, especially in winter and spring. For summer the weighting leads to an increasing significance of R2-PC that becomes equally likely as R1-NPC. The reservoir category occurs only in summer and autumn, and is negligible in other seasons.

Shifting today's FAD is tested as a contrail mitigation technique. Shifting flights 0.8 km higher reduces contrail formation in winter, while a reduction in flight altitude during summer is required to minimize potential contrail formation. Nevertheless, maximum deviations in either direction are limited by increasing air density and aerodynamic drag (lower boundary) as well as flying into the stratosphere (upper boundary) which may present other drawbacks.

The RS profiles were further examined regarding linkages with the thermal tropopause (TT) and the jet stream (defined as the altitude of maximum wind speed). The median altitude of R1-NPC is located at the TT (summer) and up to 1.5 km above the TT (winter). R1-NPC are located between $-2$ km (winter) and $-1$ km below the TT. With respect to the jet stream, the median altitude of R1-NPC is 2 km (winter) and 1 km (summer) above the jet stream. R2-PC are identified to be at the same altitude as the jet stream also following the interannual variation in jet stream location.

Considering prospective engine developments, we analyzed the influence of an increase in propulsion efficiency $\eta$ on potential contrail formation. It is found that an increases in $\eta$ from 0.3 to 0.4 leads to a general increase in potential contrail formation, particularly in R1-NPC ranging from 9 % (winter) to 26 % (summer). In connection with the further development of propulsion systems, the use of alternative fuels like ethanol, methane, and hydrogen is an option and the implications on potential contrail occurrence are estimated. It is assumed that hydrogen is burned in engines comparable with today's technology, rather than used in a fuel cell. We estimated the influence of these fuels on the likelihood of potential contrail formation. Switching to either of the alternative fuels leads to a general increase in potential contrails, again particularly of R1-NPC. The largest increase was found for hydrogen with an increase of 155 % in summer. For ethanol and methane an increase in R1-NPC of 87 % and 94 % was identified, respectively. For R2-PC the increase is less significant. Switching to hydrogen would increase the number of R2-PC by up to 31 % in summer. For ethanol and methane the maximum increase in R2-PC is found in summer with around 23 % for both fuels. It has to be emphasized that the primary objective is to minimize or entirely avoid PC as they are suspected to cause the major fraction of contrail related radiative forcing, while the effect of NPC is almost negligible (Kärcher, 2018; Teoh et al., 2020a).

These results may not generalize to other regions and other latitudes. It will be important to repeat such an analysis with a larger number of RS climatologies. It may also be interesting to combine changes in $\eta$, fuel type and flight altitude distribution to better represent what a future fleet may look like.

## Appendix A: Post-processing and corrections of radiosonde data

Radiosonde profiles are subject to biases from multiple sources. Two main effects are the direct solar illumination of the sensors and the sensor inertia, which must be corrected for. The applied post-processing is described in the following.

The first correction compensates the impact of direct solar radiation on the RH sensor. The induced artificial heating increases the RH sensor temperature with respect to the ambient temperature and leads to a dry bias in the recorded RH (Miloshevich et al., 2004). The heating and the dry bias become even more pronounced with altitude as the air density decreases and the heat

conduction between the sensor and the surrounding air is reduced. Furthermore, sensor heating intensifies with altitude as the remaining atmosphere above the RS absorbs less and less of the incoming radiation. The heating effect further depends on the sensor size and orientation towards the Sun.

Leiterer et al. (1997) and Dirksen et al. (2014) proposed RH dry-bias correction methods, which partly rely on radiative transfer simulations (RTS) to estimate the heating effect. Instead of estimating the artificial heating with RTS, the M10 RS

provides direct measurements of RH sensor temperature $T_{\mathrm{RS,RH}}$. The dedicated temperature sensor for $T_{\mathrm{RS}}$ is smaller than the RH sensor (8x9 mm in size) and is protected with an aluminum coating, reflecting 95 % of the shortwave and longwave radiation. It is assumed that $T_{\mathrm{RS}}$ therefore represents the 'true' ambient temperature and is regarded as the reference. The deviation between $T_{\mathrm{RS,RH}}$ and $T_{\mathrm{RS}}$ is used to remove the RH dry bias.

The corrected RH $RH_{\mathrm{RS,cor}}$ is determined by:

$$RH_{\mathrm{RS,cor}} = RH_{\mathrm{RS}} \cdot \frac{e_{\mathrm{sat}}(T_{\mathrm{RS,RH}})}{e_{\mathrm{esat}}(T_{\mathrm{RS}})}, \tag{A1}$$

where $RH_{\mathrm{RS}}$ is the biased RH and $e_{\mathrm{sat}}(T_{\mathrm{RS,RH}})$ and $e_{\mathrm{sat}}(T_{\mathrm{RS}})$ are the saturation water vapor pressure at the temperature of the RH sensor and the ambient air, respectively. The saturation pressure $e_{\mathrm{sat}}$ is calculated with respect to a plane liquid water surface.

The second correction of the RH measurements addresses the time-lag of the RH sensor. With decreasing temperature the

575 diffusion of water molecules into and out of the capacitor's substrate is reduced. The time response of a sensor is quantified by the time constant $\tau$, which is the required time to reach 63 % ($\approx 1 - 1/e$) of the signal caused by an instantaneous change of the ambient conditions. The time constant $\tau$ is temperature dependent. $\tau(T)$ is determined by laboratory experiments performed by the RH sensor manufacturer as well as from Dupont et al. (2021). The experiments cover a temperature range from $-70$ to $-20^\circ$C. The measurements of the response time were fitted with an exponential function:

$$\tau = A \cdot \exp(b \cdot T_{\mathrm{RS,RH}}) \tag{A2}$$

with the sensor temperature $T_{\mathrm{RS,RH}}$ (in $^\circ$C) and the fitting parameters $A = 1.3038$ and $b = -0.07002$.

The RH measurements are time-lag corrected similar to Wang et al. (2002) and Miloshevich et al. (2004). Following Miloshevich et al. (2004) the time-lag and dry-bias corrected $RH_{\mathrm{RS,tl}}$ is determined by:

$$RH_{\mathrm{RS,tl}}(t) = \frac{RH_{\mathrm{RS,cor}}(t) - RH_{\mathrm{RS,cor}}(t-1) \cdot X}{1 - X}, \tag{A3}$$

with $X = e^{-\Delta t/\tau}$, and $RH_{\mathrm{RS,cor}}(t)$ and $RH_{\mathrm{RS,cor}}(t-1)$ are the measured, dry-bias corrected RH at their respective time-steps $t$ and $t-1$. The M10 data is available with 1 Hz resolution ($\Delta t = 1\,\mathrm{s}$).

The time-lag correction is sensitive to instantaneous changes in $RH_{\mathrm{RS,cor}}$, especially with increasing altitude and increasing $\tau$. Small variations in the dry-bias corrected $RH_{\mathrm{RS,cor}}$ between two time-steps must be driven by a large change in the ambient

RH. Therefore, the correction of $RH_\mathrm{RS,cor}$ amplifies noise that is present in the raw profiles of $RH_\mathrm{RS,cor}$, $T_\mathrm{RS,RH}$, and $T_\mathrm{RS}$.
To remove the noise, a box-car filter over 20 time-steps is applied before correcting with Eq. A3. The smoothing window is selected as a compromise of noise reduction and preservation of the original signal shape. During the iterative time-lag correction, the signal is checked for plausibility. It is assumed that the ambient conditions of $RH_\mathrm{RS,cor}$ do not change by more then $\pm 0.3\,\%$ between individual measurements ($\Delta t = 1$ s, approx. 5–8 m distance). If the difference of RH $RH_\mathrm{RS,cor}(t)$ from the current and $RH_\mathrm{RS,cor}(t-1)$ of the previous time-step differ by more than $\pm 0.3\,\%$, the maximum value of $\pm 0.3\,\%$ (upper
allowed boundary) is assigned.

For testing purposes, the above outlined corrections are applied to RS profiles from the Trappes station, where RS observations with the M10 RS are performed continuously since 2012. While from 2012 to mid-2018 only limited post-processing of the RS data was applied, routine post-processing following the GCOS Reference Upper-Air Network (GRUAN Dirksen et al., 2014) specifications is available since mid-2018. As GRUAN is regarded as the reference quality standard for RS, the
GRUAN-corrected profiles provide a reference to test the previously described corrections.

The test is applied to RS from May 2021. Even though the month of May 2021 is out of the analyzed eight-year time period, it is assumed that the corrections are consistent back in time, as the same radiosonde type M10 was operated. The month of May provides a suitable test case to estimate errors caused by solar heating as the sun reaches intermediate solar zenith angles, mostly affecting the RH sensor by slant illumination.

Figures A1a and b show monthly mean vertical temperature profiles $T_\mathrm{RS}$ (blue) measured by the radiosonde and the RH temperature sensor $T_\mathrm{RS,RH}$ (green) for night-time and day-time profiles. As a reference, the GRUAN-corrected, vertical temperature profiles $T_\mathrm{RS,GRUAN}$ is given in black. Figure A1c shows vertical profiles of absolute difference in $T$. For the night-time profiles (Fig. A1a), the differences between $T_\mathrm{RS}$ and $T_\mathrm{RS,RH}$ with respect to $T_\mathrm{RS,GRUAN}$ are close to zero and overlap with the zero-line, as no solar radiation hits the sensors. It further confirms that the sensors are free of offsets. Only $T_\mathrm{RS,RH}$ shows an
increasing deviation with altitude of up to $-0.1$ K at 18 km that can be neglected.

Clear differences are found for the day-time radiosondes, which highlights the impact of solar illumination. For $T_\mathrm{RS}$ a maximum deviation of 0.8 K with respect to the GRUAN profile is identified. The differences result from the missing temperature correction that is applied during GRUAN post-processing. The discrepancies are smaller compared to the solar heating of the RH sensor, which reaches a mean absolute difference of up to 4 K at 12 km altitude.

The effects of sensor heating on the raw RH profiles are shown in Fig. A1d and e for the night-time and day-time launches, respectively. During night, RH profiles of the raw measurement $RH_\mathrm{RS}$, the corrected $RH_\mathrm{RS,tl}$, and the GRUAN-corrected $RH_\mathrm{RS,GRUAN}$ are almost identical. The maximum deviations in RH within 8–14 km altitude are $\pm 3\,\%$. Larger deviations arise in the day-time profiles, highlighting the systematic dry bias in $RH_\mathrm{RS}$ (blue). The deviations between $RH_\mathrm{RS}$ and $RH_\mathrm{RS,GRUAN}$ reach up to 13 % (between 8 and 14 km). For the corrected RH measurements $RH_\mathrm{RS,tl}$ the agreement is improved with
remaining deviations of up to 8 % and an average deviation of 1.9 % for altitudes between 8 and 14 km. The remaining average underestimation of $\pm 2\,\%$ to $\pm 3\,\%$ is attributed to the deviation between $T_\mathrm{RS}$, used in the dry-bias correction, and $T_\mathrm{RS,GRUAN}$, used in the GRUAN RH correction. During GRUAN post-processing, $T_\mathrm{RS}$ is corrected using a complex function of altitude, wind speed, and solar zenith angle. This multi-variable temperature correction was not applicable in the case of the used data.

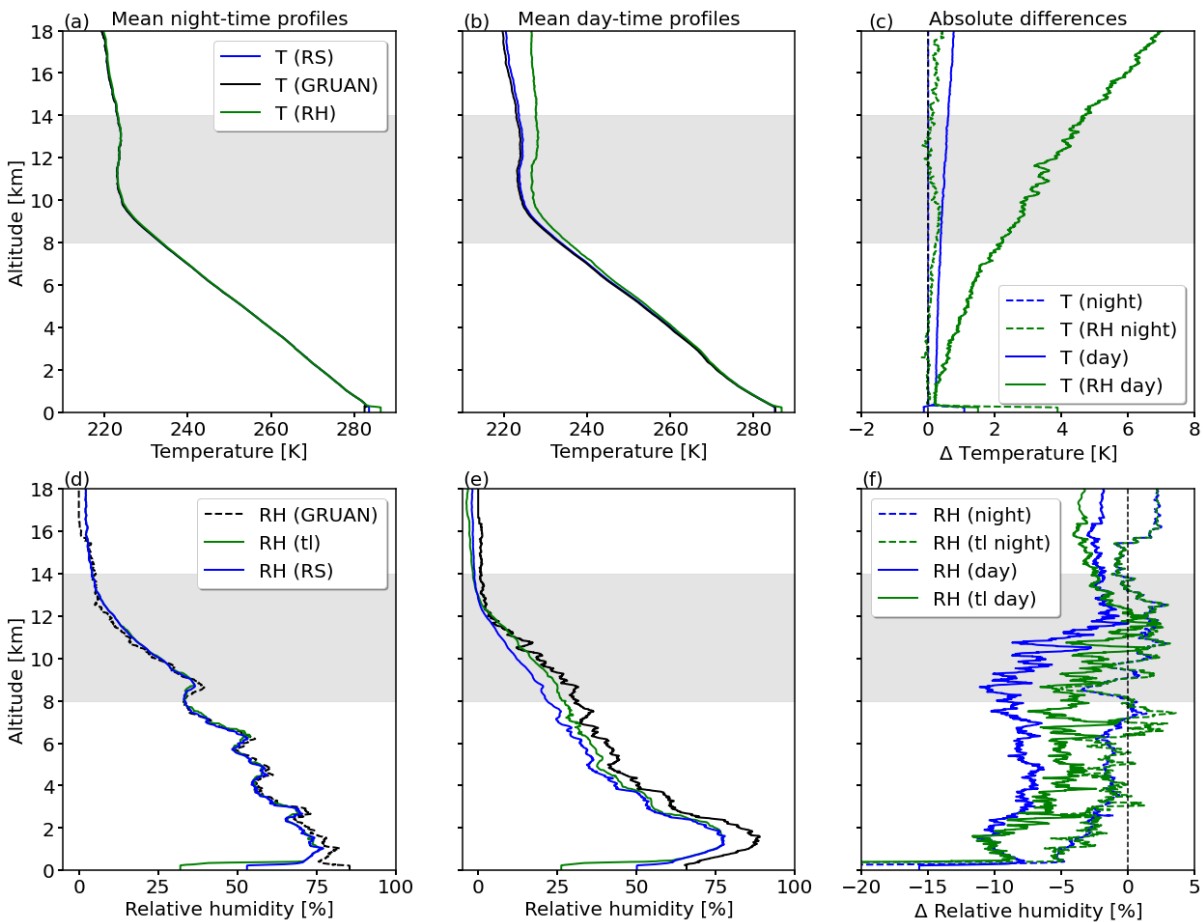

**Figure A1.** Mean profiles of uncorrected temperature (blue), GRUAN-corrected temperature profile (black), and temperature from the RH sensor (green) for **(a)** night-time and **(b)** day-time radiosondes. **(c)** Mean absolute differences of uncorrected temperature profile (blue) and the RH temperature profile (green) with respect to the GRUAN temperature profile. Day-time and night-time RS are shown with solid and dashed lines, respectively. Mean profiles of uncorrected, time-lag and dry-bias corrected, and GRUAN-corrected relative humidity for **(d)** night-time and **(e)** day-time RS. **(f)** Mean, absolute differences in relative humidity between the raw profile (blue) and the corrected profile (green) with respect to the GRUAN-reference profile. Day-time and night-time RS are shown with solid and dashed lines, respectively.

Consequently, the corrected RH measurements, analyzed in this paper, are still subject to an average dry bias of 1.9 % between 8 and 14 km.

**Appendix B: Calculation of saturation water vapor pressure over liquid water and ice surfaces**

The saturation water vapor pressure $e_{\mathrm{sat}}$ is calculated by using polynomial approximations of the Clausius–Clapeyron-relationship. Multiple approximations and equations do exist. For $e_{\mathrm{sat}}$ over liquid water $e_{\mathrm{sat,liq}}$ and ice $e_{\mathrm{sat,ice}}$ the equations after Goff and Gratch (1946); Goff (1957) are regarded as the reference, for example in Alduchov and Eskridge (1996) or Gueymard (1993). Commonly, radiosonde manufacturers use the equation after Sonntag (1994) to calculate $e_{\mathrm{sat,liq}}$. Similarly, Spichtinger et al. (2003), Immler et al. (2008), and Rädel and Shine (2010), who also analyzed radiosonde observations, used the equation after Sonntag (1994), who we follow for consistency.

After Sonntag (1994), $e_{\mathrm{sat,liq}}$ is calculated by:

$$\ln(e_{\mathrm{sat,liq}}) = \frac{c_1}{T} + c_2 + c_3 \cdot T + c_4 \cdot T^2 + c_5 \cdot \ln(T), \tag{B1}$$

with T in K and $e_{\mathrm{sat,liq}}$ in hPa. The coefficients are $c_1 = -6096.9385$, $c_2 = 16.635794$, $c_3 = -2.711193 \cdot 10^{-2}$, $c_4 = 1.673952 \cdot 10^{-5}$, $c_5 = 2.433502$. Saturation water vapor pressure $e_{\mathrm{sat,ice}}$ over ice is calculated following Murphy and Koop (2005):

$$\ln(e_{\mathrm{sat,ice}}) = c_1 + \frac{c_2}{T} + c_3 \cdot \ln(T) + c_4 \cdot T, \tag{B2}$$

with T in K and $e_{\mathrm{sat,ice}}$ in Pa. The coefficients are $c_1 = 9.550426$, $c_2 = -5723.265$, $c_3 = 3.53068$, $c_4 = -0.00728332$.

To analyze the differences in $e_{\mathrm{sat,liq}}$ as well as $e_{\mathrm{sat,ice}}$ calculated from Goff and Gratch (1946); Goff (1957), Sonntag (1994), and Murphy and Koop (2005), we compare absolute values to derive relative differences with respect to Goff and Gratch (1946); Goff (1957). For $e_{\mathrm{sat,liq}}$ the largest relative differences, over the temperature range from $-70$ to $+30°$C, are found for Sonntag (1994) with up to 4 % in $e_{\mathrm{sat,liq}}$ for the extreme case of $-70°$C. For $e_{\mathrm{sat,ice}}$ the relative differences among the approximations are below 0.2 % over the temperature range from $-100$ to $0°$C. Based on these differences in $e_{\mathrm{sat}}$ we argue that the selected approximations for $e_{\mathrm{sat}}$ are well below the measurement uncertainties and are negligible.

**A1**

**Table A1.** Notations

| Symbol | Long-name | Unit |
|---|---|---|
| $c_\mathrm{p}$ | Isobaric heat capacity | $\mathrm{J\,kg^{-1}\,K^{-1}}$ |
| $e_\mathrm{sat,liq}(T)$ | Saturation water vapor pressure over liquid water | Pa |
| $e_\mathrm{sat,ice}(T)$ | Saturation water vapor pressure over ice | Pa |
| $\eta$ | Propulsion efficiency | - |
| $EI$ | Water vapor emission index | - |
| $G$ | Slope of Schmidt–Appleman-criterion | $\mathrm{hPa\,K^{-1}}$ |
| $\gamma$ | Lapse rate | $\mathrm{K\,km^{-1}}$ |
| $\dot{m}_\mathrm{f}$ | Fuel rate | $\mathrm{kg\,s^{-1}}$ |
| $p_\mathrm{RS}$ | Pressure from radiosonde | hPa |
| $p_\mathrm{FAD}(z)$ | Probability distributions of flight traffic | - |
| $p_\mathrm{Rx}(z)$ | Probability distributions of contrail regions R1-NPC, R2-PC, R3-R | - |
| $Q$ | Specific heat energy | $\mathrm{J\,kg^{-1}}$ |
| $RH_\mathrm{RS,tl}$ | Time-lag corrected relative humidity from radiosonde | % |
| $RH_\mathrm{RS,liq}$ | Relative humidity from RS with respect to liquid water | % |
| $RH_\mathrm{RS,ice}$ | Relative humidity from RS with respect to liquid water | % |
| $RH_\mathrm{liq}$ | Relative humidity with respect to liquid water | % |
| $RH_\mathrm{ice}$ | Relative humidity with respect to ice | % |
| $RH_\mathrm{ice,crit}$ | Critical relative humidity threshold for ice-supersaturation | % |
| $RH_\mathrm{crit}$ | Critical relative humidity from Schmidt–Appleman-criterion | % |
| $RH_\mathrm{RS,cor}$ | Relative humidity from radiosonde after correction for sensor dry bias | % |
| $\tau$ | Time-constant of relative humidity sensor | s |
| $T$ | Temperature | K |
| $T_\mathrm{crit}$ | Critical temperature provided by Schmidt–Appleman-criterion | K |
| $T_\mathrm{RS}$ | Temperature from radiosonde | K |
| $T_\mathrm{RS,RH}$ | Temperature from radiosonde relative humidity sensor | K |
| $v$ | Aircraft speed | $\mathrm{m\,s^{-1}}$ |

*Code and data availability.* The primary python code - including routines, the pre-processed flight altitude distributions, and the calculated vertical profiles of contrail occurrence- are available in the supplementary data set. Further data is available on individual request.

*Author contributions.* **KW** was responsible for the post-processing of the radiosonde data, the analysis, and the preparation of the manuscript. **NB** and **OB** contributed equally to the preparation of the manuscript.

*Competing interests.* The authors declare that they have no conflict of interest.

*Acknowledgements.* The authors acknowledge support from the Direction Générale de l'Aviation Civile through the Convention N°2021-39 relative to "Aviation & Climate". This study benefited from the ESPRI computing and data centre (https://mesocentre.ipsl.fr) which is supported by CNRS, Sorbonne Université, Ecole Polytechnique and CNES, as well as through national and international grants.

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
