# Peer review of "Long-term upper-troposphere climatology of potential contrail occurrence over the Paris area derived from radiosonde observations"

_Atmospheric Chemistry and Physics, 2022_

## Author Comment (AC1)

For better legibility, the Reviewer's comments are highlighted in **bold** and changes in the manuscript are in *italic*.
* * *
**Major Comments**

**[Sections 4.1 and 4.4] Are these results (i.e., ISSR properties, potential contrail formation, and their respective seasonal variabilities) consistent with existing publications that used radiosonde measurements and in-situ water-vapour measurements from aircraft1–3? It would be great if the authors can compare and quantify the differences in their results relative to existing studies and include a short discussion on the potential reasons causing these discrepancies.**

This comment is helpful. Including previous studies and their analysis helps to set the results of this study in the scientific context. Therefore, the paragraphs in sections 4.1 and 4.4 were extended.

> *Generally lower frequencies of occurrence are detected for R2-PC conditions (PC, blue curve) with peak values of 33% during summer and 24% in spring. Autumn is characterized by an intermediate peak probability of 31% and for winter a maximum of around 28 % is identified. The annual cycle in R2-PC conditions is less pronounced compared to the R1-NPC conditions. The largest vertically-integrated occurrence is in winter followed by spring, autumn, and summer, similarly to R1-NPC conditions. This is comparable to Petzold et al. (2020) who analyzed 5 years of Measurement of Ozone and Water Vapor by Airbus In-Service Aircraft (MOZAIC Marenco et al., 1998) observations. Petzold et al. (2020) found a minimum in ISSR, a prerequisite for R2-PC, of 20 % to 30 % in summer and 35 % to 40 % in winter. Nevertheless, Petzold et al. (2020) observed a stronger seasonal dependency compared to our analysis. In contrast, much lower occurrences were found by Rädel and Shine (2007), with only 10 % of the winter profiles and 17 % of the summer profiles containing R2-PC conditions. The difference between our study and Rädel and Shine (2007) might be explained by the filtering of the radiosonde data. Rädel and Shine (2007) used only measurements below the tropopause and rejected profiles with RHliq larger than 90 % to remove conditions with iced sensors. The filtering might bias the results towards lower RH profiles. Even lower frequency of occurrence are given for by a recent paper from Agarwal et al. (2022), who found R2-PC conditions in only 3 % to 6 % of the profiles launched between 30N and 60N. The vertical position of the peaks of the R2-PC conditions are located between 10 and 11.5 km, which is around one kilometer higher compared to Rädel and Shine (2007), who estimated the mean altitude of ISSR between 9.5 and 10 km in winter and summer, respectively. Similar altitudes like in the present paper are reported by Agarwal et al. (2022), with around 9 km in winter and 11 km in summer.*
>
> *[…]*

*The vertically integrated values of contrail–weighted R2-PC occurrence ranges between 15 % and 22 %. This is significantly higher compared to Agarwal et al. (2022), who identified only 3% to 5 % of the fuel-burn weighted profiles as suitable for persistent contrails. Potential explanations for the differences are the data-sets for weighting the profiles and the overall lower occurrence of PC in the Agarwal et al. (2022) analysis. The differences in frequency of occurrence and mean altitude of R2-PC conditions between this study, Petzold et al. (2020), and Agarwal et al. (2022) likely result from the deviating sampling regions. While the present study uses radiosondes from one station, Petzold et al. (2020) uses observations that cover Europe, the North Atlantic, and North America, and Agarwal et al. (2022) uses selected radiosonde observations between 30◦ N and 60◦N. Furthermore, the reliability and accuracy of radiosonde observations near and above the tropopause are difficult and require careful post-processing. The methods and filtering during the post-processing process can influence the results and explain the differences among our analysis, Rädel and Shine (2007), and Agarwal et al. (2022).*

**[Section 4.6] The authors estimate the change in the non-persistent and persistent contrail formation from different fuel types. However, as pointed out, these fuels have differences in the water vapour emissions index and nvPM number emissions index, both of which are expected to change the various contrail properties such as the lifetime and radiative forcing4–6. While I understand that the changes in contrail properties are beyond the scope of this study, it would still be appropriate to add a short discussion on the potential changes in contrail properties due to the use of different fuel types.**

The Reviewer is right and points out an important topic. The following paragraph was added to Section 4.6 to highlight the dependence of contrail formation and contrail properties on the amount of emitted soot particles, as the soot particle number concentration is likely to decrease when switching to alternative fuels like hydrogen.

*In the previous analysis one important aspect of contrail and cirrus formation is neglected. Switching to alternative fuel types is accompanied with a suggested reduction of the soot particle number concentration in the exhaust plume and the number of available ice condensation nuclei. Kärcher (2018) showed that for soot-rich regimes ($N \approx 10^{14} - 10^{16}$ $kg^{-1}$), an increase in the particle number concentration leads to a linear increase in the number of activated ice crystals. Unexpectedly, a transition to a soot-poor regime ($N \approx 10^{12} - 10^{14} kg^{-1}$ ) leads to an increase of activated ice condensation particles. The increase in the soot-poor regime is explained by the formation and activation of aqueaous particles, which subsequently freeze, when the ambient air is sufficiently cold and humid (Kärcher, 2018) . Case studies from Kärcher et al. (2015); Kärcher and Voigt (2017) as well as measurements from Moore et al. (2017) and Voigt et al. (2021) provide support to the model from Kärcher (2018) although there remain large uncertainties. In addition, a recent study by Bier et al. (2022) showed that switching to bio-fuels also modifies the size of the ice particles, which further modifies the optical properties, hence the radiative forcing, and the contrail life-time.*

**[Section 4.7] The results in this section were very well described. However, the authors can improve it further by highlighting the potential factors that contribute to the seasonal difference in flight altitude changes in minimising persistent contrail formation. For example, it would be great if these results can be related to the seasonal cycle of the thermal tropopause height, as illustrated in Figure 5.**

Following the suggestion of the Reviewer, an additional paragraph was added to provide a link between the annual cycle of R2-PC conditions and the flight altitude deviations.

> *As shown in Fig. 5 the region for R2-PC is subject to an annual cycle, which is highest in summer and lowest during the winter month. In summer the overlap with today's FAD is largest, with the median of R2-PC and FAD at similar altitude. Therefore, flying at higher or lower altitudes reduces the vertically integrated chance for R2-PC formation. During the winter months, when the R2-PC is generally lower, a shift to higher flight altitudes reduces the overlap. Similarly, the lower median altitude of R2-PC during winter explains why higher cruising altitudes reduce the vertically integrated chance for R2-PC.*

**Minor Comments**

**[Line 43] It would be great to add a short sentence describing the second-order effects of contrails in affecting the natural cirrus properties7,8.**

We thank the Reviewer for pointing this out. This is an important factor with respect to the total cloud cover at cirrus altitude. Accordingly, the paragraph was extended.

> *However, it is still unclear to which extent contrails alter the occurrence of natural cirrus. Contrails modify the water vapor budget around them, leading to a competition for available water vapor supersaturation through condensation on ice particles (Ponater et al., 2021). This can lead to reduced natural cloud cover, a change of cloud optical properties, as well as the lifetime of natural cirrus (Burkhardt and Kärcher, 2011; Ponater et al., 2021).*

**[Line 53] The terms "terrestrial RF" and "solar RF" that were used in this paper are not commonly used in the literature. The authors can consider renaming them to "shortwave (SW)" and "longwave (LW)" RF here and in other parts of the paper to conform to the terminology that is used in the literature.**

Following the suggestion from the Reviewer, the terms "terrestrial" and "solar" are exchanged for "longwave" and "shortwave", respectively. The changes are not copied into this reply but we direct the Reviewer to the difference file of the manuscript.

**[Line 272] There is a typo where "R1-NOC" should be "R1-NPC".**

The typo is corrected and 'R1-NOC' is replaced by 'R1-NPC' .

**[Lines 430 - 432] This statement conflicts with Figure 5c. The stratosphere is generally dry and identified where the background H2O is below 10 ppm. Ice supersaturation in the stratosphere is a rare event and Figure 5c shows that persistent contrails generally form below the troposphere.**

We partly agree with the comment of the Reviewer and rephrased the section of the text, highlighting that supersaturation in the lower stratosphere is a rare event. Nevertheless, such events have been observed[1]. Furthermore, multiple definitions for troposphere and stratosphere exist. The one mentioned by the Reviewer refers to the definition by water vapor concentration. Alternatively, a definition by temperature is used[2].

> *[...]Furthermore, flying higher may have other drawbacks, especially if the fraction of flights cruising in the stratosphere increases. Emission of water vapor in the lower stratosphere has a stronger radiative effect compared to emissions in the tropopause. In addition, Gierens et al. (1999) found that, on some occasions, supersaturation can be present in the lower most part of the stratosphere and contrails that form in the lower stratosphere may have a longer life-time given the larger stratification compared to the troposphere (Gierens et al., 1999; Irvine et al., 2012). While supersaturation in the stratosphere might be rare, the increase in total flights in the lower stratosphere combined with the extended life-time can lead to an increased contrail coverage. It is also important to recognize that these results may not generalize to other sites and to other latitudes. A similar study would be needed with a much larger set of RS locations before robust conclusions can be reached. [...]*

**[Lines 477 – 485] It would be a great for the authors to highlight that the contrail climate forcing from NPC are generally negligible, and that PC's are the significant contributor to contrail climate forcing.**

The Reviewer is right and this is an important comment to interpret the results and to give advice on which contrail type should be avoided. The following text was added to the paragraph to highlight the importance of PC mitigation.

> *It has to be emphasized that the primary objective is to minimize or entirely avoid PC as they are suspected to cause the major fraction of contrail related radiative forcing, while the effect of NPC is almost negligible (Kärcher, 2018; Teoh et al., 2020a).*

[1] (Gierens et al., A distribution law for relative humidity in the upper troposphere and lower stratosphere derived from three years of MOZAIC measurements, 1999, Ann. Geophys. , Vol. 17, No. 9, p. 1218-1226)
[2]  (Petzold et al., 2020, Ice-supersaturated air masses in the northern mid-latitudes from regular in situ observations by passenger aircraft: vertical distribution, seasonality and tropospheric fingerprint, 2020, Atmos. Chem. Phys. , Vol. 20, No. 13, p. 8157-8179)

---

## Author Comment (AC2)

For better legibility, the Reviewer comments are highlighted in **bold** and changes in the manuscript are in *italic*.
* * *
**Comments**

**Is there a reason that for the calculations of the saturation water vapor pressure over liquid water the equation after Sonntag (1994) is used? Wouldn't it be more consistent to use the Murphy and Koop (2005) method also for e_sat_liq (as it is already done fore_sat_ice)?**

The saturation water vapor pressure $e_{sat}$ is calculated by using polynomial approximations of the Clausius–Clapeyron-relationship. Multiple approximations and equations do exist. For $e_{sat}$ over liquid water $e_{sat,liq}$ and ice $e_{sat,ice}$ the equations after Goff and Gratch (1946); Goff (1957) are regarded as the reference, for example in Alduchov and Eskridge (1996) or Gueymard (1993). Commonly, radiosonde manufacturers use the equation after Sonntag (1994) to calculate $e_{sat,liq}$. Similarly, Spichtinger et al. (2003), Immler et al. (2008), and Rädel and Shine (2010), who also analyzed radiosonde observations, used the equation after Sonntag, which we follow for consistency.

To analyze the differences in $e_{sat,liq}$ as well as $e_{sat,ice}$ calculated from Goff and Gratch, Sonntag, and Murphy and Koop, we compare absolute values to derive relative differences with respect to Goff and Gratch.

[Figure]

The first figure below shows (a) the calculated saturation $e_{sat,liq}$ with respect to liquid water using the equations of Goff and Gratch (black), Sonntag (green), and Murphy and Koop (blue). Sub-panel (b) shows the ratio between Sonntag (green) or Murphy and Koop (blue) with respect to Goff and Gratch. Over the plotted temperature range from -70 to 30° C the differences are of at most 4% for Sonntag in the extreme case of -70° Celcius.

[Figure]

The second figure shows (a) the calculated supersaturation with respect to ice using the equations of Goff-Gratch (black), Sonntag (green), and Murphy and Koop (blue). Sub-panel (b) shows the ratio between Sonntag (green) or Murphy and Koop (blue) with respect to Goff-Gratch. Over the plotted temperature range from -100 to 0° C the differences are below 0.2%. Based on these minor differences in calculated supersaturation we argue that the selected approximations for the supersaturation do not influence the results over the temperature range applied here.

A part of this discussion is added to the Appendix to the manuscript as it could be useful for the interested reader.

**In order to help readers better understand the results, a rough quantitative estimate of the possible disadvantages of higher / lower altitudes would be helpful.**

The Reviewer points to an important topic, which has to be considered for flight planing and avoiding ISSR. Though it is beyond the scope and capability of this study to provide quantitative estimates. The topic is complex and dedicated studies are ongoing to determine and quantify the effects of flying at lower or higher altitudes. Nevertheless, we added the following paragraph to the manuscript to point the interested reader to already existing studies.

> *[…] One has also to consider that aircraft tend to fly at / or close to their optimal flight levels. Flying lower or higher may be sub-optimal in terms of fuel consumption and/or require adjusting the aircraft airspeed. Furthermore, depending on an aircraft characteristics and*

*payload, it may not always be possible to fly higher. Beyond that the interactions between flight altitude, aircraft performance, engine emissions, and radiative impact of possible contrails are complex. To estimate the effect of flight altitude changes dedicated studies have been performed, for example by Frömming et al. (2012), Dahlmann et al. (2016), and, more recently, by Matthes et al. (2021).*

Furthermore, a quantification is not possible as we a missing the import point of the ice crystal size after the contrail formation process. In case of hydrogen engines, chemistry calculations would be required.

**The illustrations are very clear and meaningful. Nevertheless, an indication of the percentiles for the tropopause (Fig. 5) and jet stream height (Fig. 6) would be desirable in order to gain an impression of their variability.**

Following the suggestion of the Reviewer we added lines for the 20, 40, 60, and 80th percentile. The revised plots are shown below and exchanged in the manuscript.

[Figure]

**P11L258: „Profiles for which the temperature inversion was weak and the TT altitude was not clearly identifiable are removed from the analysis." What exactly do you mean**

**with „weak"? How do you define this „weakness"?**

We agree with the Reviewer that 'weak' is an insufficient description. Therefore, the paragraph has been rephrased to explicitly describe the selection algorithm.

> *[…] For each RS profile, γ is calculated with Eq. 9 and the location of the smallest γ (in absolute value), i.e., the local minimum in T , between 8 and 14 km is set as the TT. For profiles, where the altitude of smallest γ was equal to 8 or 14 km, the TT altitude was not identifiable and was removed from the analysis.*

**P14L332: "The smallest distances are identified in March and November with the jet stream at the same altitude as for R2." For me it seems that the distance in April is smaller than the distance in March. Typo?**

This is correct. The distance is smallest for April.  The typo was corrected.

> *The smallest distances are identified for April and November with the jet stream at the same altitude as for R2-PC*

**P12 L278 "similarly to R2-PC conditions" - you mean R1-NPC? Typo?**

The Reviewer is right. We refer to the R1-NPC. The typo was corrected.

> *The largest vertically-integrated occurrence is in winter followed by spring, autumn, and summer, similarly to R1-NPC conditions.*

**P14L331: "Similarly Fig 6b..." you mean Fig 6c?**

The reviewer's statement is correct and the typo was corrected.

> *Similarly, Fig. 6c visualizes the vertical distribution for R2-PC.*

**Minor Comments**

**P12 L273. R1-NOC should be R1-NPC**

The typo is corrected and 'R1-NOC' is replaced by 'R1-NPC' .

**Sometimes R2 is used, other times R2-PC (same for R1 and R1-NPC). Please use consistent labeling in the text and graphics.**

The nomenclature has been homogenized throughout the paper. Please see the diff file.

**References used in the author answers:**

Alduchov, O. A. and Eskridge, R. E.: Improved Magnus Form Approximation of Saturation Vapor Pressure, J. Appl. Meteorol., 35, 601 – 609, https://doi.org/10.1175/1520 0450(1996)035<0601:IMFAOS>2.0.CO;2, 1996.

Gueymard, C.: Assessment of the Accuracy and Computing Speed of Simplified Saturation Vapor Equations Using a New Reference Dataset, J. App. Meteorol., 32, 1294 – 1300, https://doi.org/10.1175/1520-0450(1993)032<1294:AOTAAC>2.0.CO;2, 1993.

Immler, F., Treffeisen, R., Engelbart, D., Krüger, K., and Schrems, O.: Cirrus, contrails, and ice supersaturated regions in high pressure systems at northern mid latitudes, Atmos. Chem. Phys., 8, 1689–1699, https://doi.org/10.5194/acp-8-1689-2008, 2008

Rädel, G. and Shine, K. P.: Evaluation of the use of radiosonde humidity data to predict the occurrence of persistent contrails, Q. J. Roy. Meteor. Soc., 133, 1413–1423, https://doi.org/10.1002/qj.128, 2007

Spichtinger, P., Gierens, K., Leiterer, U., and Dier, H.: Ice supersaturation in the tropopause region over Lindenberg, Germany, Meteorologische Zeitschrift, 12, 143–157, https://doi.org/10.1127/0941-948/2003/0012-0143, 2003